# Individual Differences in Bodily Self-Consciousness and Its Neural Basis

**DOI:** 10.3390/brainsci14080795

**Published:** 2024-08-08

**Authors:** Haiyan Wu, Ying Huang, Pengmin Qin, Hang Wu

**Affiliations:** 1Key Laboratory of Brain, Cognition and Education Sciences, Ministry of Education, School of Psychology, Center for Studies of Psychological Application, Guangdong Key Laboratory of Mental Health and Cognitive Science, South China Normal University, Guangzhou 510631, China; 20212921003@m.scnu.edu.cn (H.W.); ying_huang@m.scnu.edu.cn (Y.H.); 2Pazhou Lab, Guangzhou 510330, China; 3Key Laboratory of Brain, Cognition and Education Sciences, Ministry of Education, Institute for Brain Research and Rehabilitation, Guangdong Key Laboratory of Mental Health and Cognitive Science, South China Normal University, Guangzhou 510631, China

**Keywords:** bodily self-consciousness, individual differences, self-processing, neural basis, healthy subjects

## Abstract

Bodily self-consciousness (BSC), a subject of interdisciplinary interest, refers to the awareness of one’s bodily states. Previous studies have noted the existence of individual differences in BSC, while neglecting the underlying factors and neural basis of such individual differences. Considering that BSC relied on integration from both internal and external self-relevant information, we here review previous findings on individual differences in BSC through a three-level-self model, which includes interoceptive, exteroceptive, and mental self-processing. The data show that cross-level factors influenced individual differences in BSC, involving internal bodily signal perceptibility, multisensory processing principles, personal traits shaped by environment, and interaction modes that integrate multiple levels of self-processing. Furthermore, in interoceptive processing, regions like the anterior cingulate cortex and insula show correlations with different perceptions of internal sensations. For exteroception, the parietal lobe integrates sensory inputs, coordinating various BSC responses. Mental self-processing modulates differences in BSC through areas like the medial prefrontal cortex. For interactions between multiple levels of self-processing, regions like the intraparietal sulcus involve individual differences in BSC. We propose that diverse experiences of BSC can be attributed to different levels of self-processing, which moderates one’s perception of their body. Overall, considering individual differences in BSC is worth amalgamating diverse methodologies for the diagnosis and treatment of some diseases.

## 1. Introduction

The self has long been employed in the research of psychology and neuroscience, whose integration can be perceived as extending from the internal body to the external natural and social environment [1,2]. Self-consciousness is closely connected with the self and regarded as the feeling of self-experience, which relies on the brain’s dynamic processing of diverse bodily signals [3]. Bodily self-consciousness (BSC) is a fundamental aspect of human experience that refers to the subjective awareness of the body as an aspect of the self-conscious experience. It has increasingly become the focus in the field of neuroscience [4,5]. BSC encompasses the complex integration and encoding of various bodily signals, enabling the experience of distinct physical sensations [6]. Researchers have thoroughly investigated the construction of BSC through the paradigms of body illusion, such as the rubber hand illusion (RHI) and the whole-body illusion [6,7,8,9]. It was found that BSC stemmed from the multisensory integration and varied bodily experiences (including body ownership, self-location, first-person perspective, and agency) [10,11]. Although the construct of our bodily self is generally perceived as a stable aspect of conscious experience, the presence of individual differences in BSC has been established [12,13]. These differences reflect horizontal variance in personal traits, including personality, cognitive capabilities, age, and early life experiences [12,14,15,16]. For example, experiments utilizing the RHI paradigm have revealed that factors like age and the width of the temporal binding window (i.e., the time frame within which multisensory signals are either integrated or segregated) could affect multisensory integration for body representation [12,13,17].

Previous studies have suggested that there are multiple levels of self-processing, which refers to the processing of self-related body-environment information. In the context of self-processing, the perception of self-relevance needs to be progressive from low-level (interoception and exteroception) to high-level (mental self) features [1,18,19]. Thus, a multilevel model can provide different perspectives to understand the individual differences in BSC. For instance, the decline in BSC perception level (such as inaccurate heartbeat perception) and some special diseases (such as body agnosias) reflect the decline in integration between different levels of self-processing. BSC needs to integrate self-information effectively at different levels. Accordingly, our review is summarized from interoceptive, exteroceptive, and mental perspectives according to the experimental paradigm and stimulus, so as to understand the individual differences in BSC more systematically. Notably, our review is based on our previous theoretical model, which holds that there are three levels of self-processing, including interoceptive, exteroceptive, and mental self-processing [2]. The three-level-self model is used to explain how the brain integrates internal and external environmental information to reveal the relationship between body, environment, and mental state in the self. The model embodies the mechanism of how external stimuli become self-related and thus integrate into the self, thereby extending the self beyond the physical body to encompass the environment. In addition, the results show that the common involvement of the insula was found across all three levels of self-processing, which indicates that the integration of internal sensory signals may be the key to self-processing. In addition, cingulate cortex and anteromedial prefrontal cortex (amPFC) may play a role in combining non-physical external stimuli with body signals to form self-related pathways. The temporoparietal junction (TPJ) plays an important role in integrating multisensory input and motion perception signals. These findings provide insights into how the brain processes complex information related to BSC.

Individual differences in BSC can be attributed to various aspects [20,21]. Previous opinions explaining the causes of the individual differences have primarily focused on how the brain distinctively processes multisensory signals (e.g., visual, tactile, and proprioceptive signals), resulting in unique bodily experiences [15,20,22]. However, it is important to note that multisensory input serves as merely one of many channels for obtaining information about BSC, as well as just one of the perspectives that explain individual differences [6,23]. The construct of BSC is not solely reliant on the integration of sensory cues; rather, a myriad of additional processes likely contributes to the generation of individual differences in this domain, which allows us to fully consider the aspects that cause individual differences in BSC.

The intricate relationship between our bodily experiences and the neural basis is a topic of immense interest in the field of neuroscience. Delving deeper into the diversity of BSC, the variation in BSC has piqued the curiosity of researchers, prompting a deeper investigation into its underlying neural basis. The revelation of individual differences in BSC has spurred a quest to uncover its biological roots. With the advent of neuroimaging technologies over the past decades, it has become feasible to gauge individual differences in brain structure in terms of morphology and brain function [1,24,25]. Utilizing these advancements, we can now appraise the brain structure linked to the diversity of personal traits, such as estimating local gray matter volume and cortical thickness [26,27,28]. Furthermore, the examination of the variations in the degree of brain functional connectivity and task-based activation has also proven beneficial for probing individual differences in BSC, encompassing the neural correlates of the first-person perspective [29]. We will discuss the neural basis of individual differences in BSC according to the main brain regions involved among three levels of self-processing. Specifically, the insula, anterior cingulate cortex (ACC), and frontal lobe are involved in all three levels [30,31,32]. The parietal lobe and hippocampus play a role in the integration and representation of sensory information under exteroception [33,34]. The activity of the TPJ and middle temporal gyrus (MTG) is related to mental self-processing [32,35]. These main brain regions constitute the neural basis of BSC, whose variations will aid in elucidating the individual differences in BSC.

Whereas the aspects of BSC have been widely explored in the last decades, the existing literature has not systematically discussed the factors contributing to individual differences in BSC. It is necessary to highlight the importance of individual differences for the study of the bodily-self construction and consciousness, thereby enriching the current findings in the field, where the paradigms are often employed universally for groups [10,36,37]. In addition, a fundamental property of BSC is its connection to the self as the subject of bodily experiences, which is a multifaceted entity composed of both bodily and abstract non-bodily information that should be examined from distinct dimensions [1,2]. Thus, our purpose is to summarize the previous findings of individual differences in BSC from the perspective of the three-level-self model, so as to better understand the individual differences in BSC and its neural basis [2]. The model reflects how the brain transmits and integrates information related to itself from the inside or outside, involving three levels: interoceptive processing, exteroceptive processing, and mental self-processing [2]. While BSC primarily concerns interoception, it indeed necessitates dynamic interaction with information across multiple levels [38]. Hence, we will review individual differences in BSC from the perspectives of three single levels and interactions between multiple levels.

In sum, our goal is to synthesize the previous findings on individual differences in BSC from the perspective of the three-level-self model, aiming to enhance our comprehension of these individual differences and their corresponding neural basis. By meticulously examining the ways in which our bodily experiences are shaped by the specific perceptual, cognitive, and emotional elements, our review seeks to forge a deeper connection between the subjective world of self-consciousness and the objective existence of the body and environment. Centered on this goal, this review will be grounded in the three-level-self model. Specifically, we will review the research into individual differences in BSC in three steps (Figure 1). The first step is to consider individual differences in BSC that are pertinent to the single-level-of-self-processing model. The second step is to consider how interactions among various levels of self-processing influence individual differences in BSC (Table 1). The third step is to consider the neural basis, which embodies individual differences in BSC, especially in the context of brain structures in terms of morphology, brain functional connectivity, and task-based activation. This review could improve and widen the existing BSC frameworks at an individual level, as well as expose underlying causes of diseases linked to BSC. Moreover, it could pave the way for the development of targeted treatments that cater to the specific needs of individuals affected by these differences.

## 2. Individual Differences in BSC in Line with the Single-Level-of-Self-Processing Model

### 2.1. Interoceptive Processing

Interoceptive processing contributes to the construction of BSC [40], implying that individual differences in BSC may therefore be affected by individual differences in interoception. Earlier, research into interoceptive processing has highlighted the need to clearly differentiate distinct aspects of interoception, in order to formally address its dimensional issues [63,64]. Importantly, interoception is thought to encompass three dimensions: interoceptive accuracy, interoceptive sensibility, and interoceptive awareness [65,66]. Interoceptive accuracy refers to the degree of precision that allows for objective assessments of internal bodily sensations, which is also linked to emotion [67,68] and cognitive processing [69]. Interoceptive accuracy represents the objective capacity of oneself, whereas interoceptive sensibility is highly subjective, which indexes perception and belief of signals originating from the internal body [70]. As for interoceptive awareness, which represents the perception and monitoring of interoceptive accuracy, it demonstrates the correspondence between the objective physiological performance and meta-cognitive awareness [65]. Intriguingly, all three dimensions of interoception not only affect a wide range of BSC, but are also implicated in individual differences in BSC [44,62,71]. In support of this, we will review behavioral, electrophysiological, and neuroimaging studies that investigated the relationship between dimensions of interoception and BSC in the following paragraphs, to illustrate the perspective that individual differences in BSC may arise from differences in interoception.

Behavioral evidence reveals that different bodily experiences are subject to the influence of interoception. For instance, the vividness of the RHI was affected by levels of interoceptive sensibility. Specifically, contrary to people with higher interoceptive sensibility, those with lower interoceptive sensibility reported a stronger RHI, as well as a more stable perception of body ownership [43]. Additionally, Raimo et al. (2021) observed that interoceptive sensibility could also produce important effects in body representation as individuals aged. Specifically, unlike young and middle-aged adults, the children and older adults performed worse in a body schema task and body structural description task, due to the negative role that interoceptive sensibility may play in their age bracket. Equally important, interoceptive accuracy also links to individual differences in facets of BSC. For example, the peripersonal space, which refers to a proximal spatial region that extends beyond the boundaries of one’s body, serves as an interactive mediation between oneself and the surrounding environment [72]. Evidence of the association between interoceptive accuracy and peripersonal space showed that individuals with higher interoceptive accuracy possessed a narrower peripersonal space [39], as they allocated added attention resources to internal body signals, other than external ones. Moreover, self-face recognition, a hallmark of self-consciousness and sense of identity, is a paradigm that is often employed to investigate the role of interoceptive accuracy in BSC [42]. Previous studies have indicated that individuals with higher interoceptive accuracy tend to exhibit a bias in judging face stimuli as self [73]. When exposed to stimuli that relate to the bodily self, higher interoceptive accuracy may facilitate individual’s identification and sense of ownership of their own body. To the contrary, higher interoceptive accuracy tends to disrupt identification with bodily part stimuli under conditions where the stimuli may be alien to self [42]. In other words, for individuals with higher interoceptive accuracy, the self–other boundary may be more distinct. As for interoceptive awareness, previous research suggested that the interoceptive awareness of males is higher than that of females, which allows males to show greater efficiency in consciously monitoring heartbeats [74]. Moreover, individuals with higher interoceptive awareness may experience their body state with heightened accuracy [46,75], such as being less susceptible to body illusions [46].

The electrophysiological research on BSC mainly focuses on heartbeat evoked response (HER), which refers to the neural response elicited by heartbeat signals [76,77]. Research into HER had found its association with the key components of BSC, such as bodily and facial self-identification [63]. Empirical results have revealed that HER exhibits individual differences [41,78]. Additionally, HER reflects interoceptive processing, which implies that individual differences in BSC could be linked to variations in interoception [79]. Moreover, neural processing of heartbeat signals can be measured using heartbeat evoked potentials (HEP), which reveal that neural activity is primarily concentrated in regions such as the operculum, insula, cingulate cortex, and somatosensory cortex [1,80]. As with HER, the HEP, which enables us to observe dynamic bodily changes, is associated with certain aspects of BSC, such as body illusions [40]. Notably, a set of findings have indicated that the cortical activations in response to heartbeats differ among individuals [77,80]. In particular, it has been observed that individuals with higher interoceptive accuracy exhibit more pronounced HEP amplitudes [41,77,81]. As stated earlier, Park et al. (2016) discovered a positive correlation between HEP amplitudes and ratings assigned to body illusions. These findings implied a potential indirect link between individual differences in interoception and those in BSC [63].

Recent neuroimaging studies have taken the factors such as age and sex into account, concerning the association between interoception and brain regions, shedding light on the indirect connection between individual differences and differentiation in BSC. A previous study suggests that there is a positive correlation between the level of interoceptive accuracy and the connectivity of the ACC within the salience network among individuals with an average age of 77 [82]. In previous investigations, the ACC has been observed to play a significant role in various aspects of the bodily self [83,84,85]. Interestingly, among younger adults, the level of interoceptive accuracy is positively correlated with the connectivity of the insula, another important region in the salience network [30]. Similar to the function of the ACC, the intrinsic connectivity of the insula participates in the construction of BSC, such as self-identification and body ownership [86,87]. Additionally, sexual differences were observed in the influence of volumes of brain regions on interoceptive sensibility. Previous research supports the notion of a female advantage in higher interoceptive sensibility [45], potentially due to the discovery of a larger volume of the insula in females compared to males, which plays a role in interoceptive processing, as well as body representation and self-consciousness processing [45,88]. Together, the findings collectively reinforced the perspective that an indirect relationship exists between individual differences and the differentiation in BSC, supported by neuroimaging techniques.

These findings collectively make it plausible to consider that variations in interoception are intrinsically associated with the individual differences observed in BSC. Extending beyond interoception, and considering the significant impact of multisensory integration on bodily experiences, we will explore the role of exteroception subsequently.

### 2.2. Exteroceptive Processing

Based on the inputs of external sensory signals (e.g., tactile, visual, and proprioceptive), exteroception reflected the ability to perceive one’s bodily self [10]. Typically, prior research has predominantly employed the multisensory conflict paradigms (e.g., auditory–tactile, visual–tactile, visual–vestibular, etc.) to study the mechanisms of BSC from the exteroceptive perspective [89,90,91,92,93]. This is attributed to the known role that multisensory processing played a role in integrating diverse sensory signals and eliminating potential signal conflicts to maintain a coherent BSC once individuals find themselves in a situation where the sensory signals are changeable [11]. Of course, there are some abnormal examples, such as the out-of-body experience caused by the sense of disintegration of multi-sensory stimuli [6]. It is a form of consciousness that occurs outside of the physical body and may interfere with some experiences of BSC like self-location [92]. Additionally, as multisensory processes may renew, maintain, or disrupt BSC, BSC is thought to be highly plastic in the context of exteroception [11,94,95]. Notably, previous studies suggested that individual differences in BSC may relate to different principles of multisensory processing [96,97]. The key question of how the brain integrates inputs from different sensory channels and how these inputs interact with other extraneous information to shape unique self-perception of oneself will be discussed below.

The self-face representations is a critical feature of BSC, whose activation degree is related to the ability of self-face expression recognition [52]. Individual differences exist in facial expression processing ability. In specific, autism is one of the most common diseases related to poor facial expression processing. Among healthy subjects, there are obvious individual differences in autism-spectrum quotients (AQs). It was found that individuals with higher AQs were slower to recognize facial expressions [53]. In addition, research showed that the activation of self-face representation can promote emotion processing [52]. Therefore, for individuals with higher AQs, the reaction time to facial expressions under the self-face condition is significantly faster than that under the non-self-face (others’ face) condition. This discovery promotes our understanding of how one’s own body’s sensory signals, self-processing, and BSC are integrated and the distinction mechanism of self–other.

The strength of BSC is related to the degree to which an individual relies on a particular sense [98,99]. Previous evidence suggested that professional dancers, but not non-dancers, relied on proprioception more; as a result, they reported stronger subjective bodily consciousness and performed better in perceptual tasks of body posture [15]. It is suggested that professional dancers are more focused on their bodies than the external environments, which suggests that their heightened BSC may be a result of their training and the demands of the profession [98]. This observation suggests that the way in which we perceive our physical bodies and the space around us is not a uniform experience, but rather a highly individualized one. Given the role of Bayesian principles in addressing such question of sensory representation of the body, we will subsequently discuss its utilization in explaining the experience of BSC.

In the context of Bayesian causal inference (Bayesian CI) model, individual differences in BSC could indeed be affected by the way sensory signals are processed, taking into account the relative weights assigned to different senses [100]. To elucidate this view, we will illustrate the independent role of unisensory components that are involved in the inferential process [101]. Specifically, predicted by a Bayesian CI model and reliability-weighted integration principles, the brain assigns greater trust to the signals that are less noisy or more consistent [97,102]. This means that our perception is not simply a sum of all available cues but rather a dynamically weighted consequence where each sensory cue’s contribution is modulated by its reliability. Notably, sensory signals with higher relative precision tend to be more reliable, which means that sensory signals that are deemed more precise are consequently accorded greater weight in the decision-making process [103]. For instance, individuals who assign higher weights for proprioceptive signals experienced a greater sense of spatial localization of body parts and are less susceptible to body illusions [47,48], which can be attributed to their increased attention to the dependable proprioceptive information.

The Bayesian CI model could also be applied to predict the effect of multisensory processing, with the computational outcomes offering a quantitative explanation of individual differences in BSC. Utilizing the RHI paradigm, previous research has found that temporal resolution towards multisensory signals is significant in multisensory integration [49,104]. It has been widely observed that individuals with lower temporal resolution (i.e., a wider temporal binding window) of multisensory signals experience a stronger RHI [50,61]. Recent perspectives suggest that the Bayesian CI model is accepted to provide a plausible explanation of this phenomenon in the sense of quantitative computation [50,105]. The computational process of Bayesian CI is helpful for considering whether the different sensory signals stem from one common cause or not. In this evaluation, it incorporates several elements: the temporally or spatially associated features of sensory signals, the relative degree of the uncertainty of sensory signals (e.g., ambiguity that the proprioceptive signals are generated from the real body), and the empirical priors [101]. As for the RHI, it arises from the three-way interaction of vision, touch, and proprioception [11]. The Bayesian CI model considered the uncertainty of these three senses, their temporal correlation, spatial proximity, and empirical priors, to compute the strength of the RHI quantitatively in a given situation [51,106]. Accordingly, only when stimuli are presented close spatially and temporally can they be integrated, which dissected the phenomenon mentioned above about the temporal binding window. Based on these findings, Chancel et al. (2022) suggested that individual differences in the RHI can be computationally attributed to differences in the integration of bottom-up sensory cues, information regarding sensory uncertainty, and prior knowledge during the process of causal inference. The different computational results of the Bayesian CI model provide further support to the study of individual differences in BSC [47,51].

Overall, these findings indicate that investigating the processing of multisensory signals can enrich our understanding of individual differences in BSC. Moreover, they offer insights into how to take into consideration individuals’ particular perceptual traits when navigating environments with changeable sensory inputs, thereby facilitating better adaptation to such surroundings and allowing for adjusted performance within them.

### 2.3. Mental Self-Processing

Mental self-processing represents a higher level of self-processing, while interoception and exteroception are thought to be lower ones [2]. The mental self refers to the process that is closely related to the virtual and abstract sense of oneself, which tells the difference from other people [107]. It is often associated with virtual mental-level stimuli such as names, self-identity, autobiographical memories, and trait words linked to the self [108,109,110]. This level of self-processing involves brain regions like the insula, ACC, PFC, and TPJ, affecting the connection between external irrelevant information and self, which will impact the self-processing at other levels. In addition, BSC was found to be related to the mental form of self-consciousness. Bréchet (2022) found that the autobiographical self connects BSC with situational memory, allowing individuals to re-experience past scenes from the inside or outside of the body [111]. Habermas and Köber (2015) suggested that the stability and constancy of the bodily self are conducive to maintaining the coherence of self-awareness, thus supporting the diachronic continuity of personal identity [112].

Macroscopically, diverse representations of BSC may be driven by individual differences in sociocultural constructs [113]. Individuals in Western and East Asian cultures are observed to process BSC differently [16]. Particularly, in East Asian cultures, the concept of “face” is deeply rooted and refers to a person’s reputation, social status, and sense of pride or honor, which is an important aspect of interpersonal relationships and social interactions. Thus, when East Asian individuals process visual information of their own faces, they are inclined to link the observed facial features with the broader idea of “face” [16,114]. Besides, given the emphasis of perfectionism in Japanese culture, Japanese ingrain formality and normativity deeply into their self traits to the point of ignoring the actual sensation of the bodily self [54]. However, American culture is renowned for championing freedom and independence. This cultural focus on personal autonomy extends to individuals’ attentiveness to their bodily well-being, prompting them to closely monitor any internal physiological cues [55]. These findings are well aligned with what is known of the reference group theory [115], on which individuals could depend to activate BSC.

It has been demonstrated that interoceptive and exteroceptive signals can be combined with individuals’ prior knowledge and expectations to update their social identity [79,116,117,118]. Inversely, some current research found that individuals’ social identity could change the representation of BSC [56,57]. For example, for the White participants, the process of integrating visual information about a black rubber hand into their body schema was more prolonged and challenging. This was manifested in a delayed acceptance of the black rubber hand as part of their own body, as indicated by a reduced magnitude of the RHI. Additionally, there was a diminished proprioceptive drift observed for the black rubber hand [56]. Moreover, it is suggested that gender social identity affects body identification [57]. Women, due to impact of the social environment, frequently view their bodily form as a central aspect of their feminine identity [119]. Consequently, women may be more prone to develop negative self-identification regarding their bodies when compared to men [57]. These observations allow us to form deeper and meaningful connections between abstract, conceptualized self-representations and BSC [11]. To further confirm the effect of mental self-processing on BSC, more studies concerning other virtual aspects are warranted, which is a crucial step in the extensions of BSC.

In sum, these findings reveal that personal traits shaped by cultural and social factors result in distinctive perceptions of BSC. This shows that a shift from conceptualized self-representations to BSC is feasible and logical. This multicultural perspective could contribute to the development of a more universal or inclusive model of BSC.

### 2.4. Brief Summary

Altogether, each level of the self-processing model provides a precious insight into how individual differences in BSC are generated. However, the single-level-of-self-processing model struggles to accomplish the integration of oneself independently as self is a multilevel hierarchy. To maintain the integration of multilevel self-representation, there is a requisite for interaction and synchronization across diverse processing levels [2,120]. Following this section, we shall delve into how diverse levels of self-processing interact with each other to assert their pivotal influence in modulating the distinctive BSC.

## 3. Individual Differences in BSC in Line with Interactions among Diverse Levels of Self-Processing

The definition of self is still controversial. For example, there is competition and integration among different levels of self inside the body [121,122]. Consequently, this review attempts to explain the factors that are associated with the individual differences in BSC from the perspective of multilevel self-information processing [123]. Subsequently, we endeavor to investigate the interactions among the different levels of the body and aim to illuminate how the individual differences in the bodily self are achieved.

BSC stems from the integration of interoceptive and exteroceptive signals, each contributing based on its relative precision [124,125]. In this multilevel framework, given that precision is always relative, unilevel precision plays an essential role in weighting the available inputs from different levels, thereby exerting a substantial influence on the construction of BSC [125]. According to Bayesian free energy principles, the predictive coding theory linked to self-experiences necessitates the integration of “prediction” regarding one’s bodily state and “prediction error” that deviates from prior expectations [126]. This complex interplay culminates in a commitment to minimize these prediction errors, thereby optimizing the precision of the levels [58]. Within the framework of the predictive coding theory, it is postulated that the diminished interoceptive prediction errors associate with heightened interoceptive precision (interoceptive signals become more reliable) [11,127]. Ainley et al. (2016) have explored individual differences in interoceptive processing, conceiving it as individual differences in the relative precision of prediction errors. Empirical studies have shown that, for individuals with higher interoceptive precision, the prediction error of interoception is smaller, leading them to prioritize the interoceptive inputs, while simultaneously diminishing the perception of exteroceptive inputs [126,128]. Noteworthy, interoceptive accuracy is closely linked to interoceptive precision [58]. Individuals with above average interoceptive accuracy exhibited a reduced likelihood of decay in their interoceptive prediction errors and relatively diminished exteroceptive precision, resulting in lower vividness of the RHI [129] and the whole-body illusion [130].

Besides interaction among the low-level, mental self-processing, as a higher-level self-processing, is inherently associated with interoceptive processing and multisensory processing [14]. The differences of their interaction effects advance our comprehension of individual differences in BSC. Recent research on the topic of the mind are increasingly emphasizing empathy as a higher level of mental self-processing [59,131,132], which is characterized by the capacity to comprehend the mental states of others, as well as the ability to adopt their perspectives [133]. On the one hand, individual differences in empathy ability are strongly linked to interoceptive processing. In line with this perspective, there is evidence that individuals with higher empathy ability were capable of perceiving internal body signals more accurately, which is conducive to assess various aspects of the bodily consciousness [59,60]. On the other hand, different levels of empathy ability lead to the different effects of multisensory processing. For example, Asai et al. (2011) found that individuals with higher empathy ability exhibited a propensity to vividly imagine others’ perspectives by integrating visual, auditory, and other contextual cues. This propensity is positively correlated with sensitivity to the RHI. Similarly, Mul et al. (2019) noted a correlation between higher empathy ability and increased susceptibility to body illusions when exposed to synchronized visual–tactile stimuli, which exhibited stronger self-localization drift and illusory self-identification. Even more notably, empathy ability could alter BSC, potentially through enhanced integration of both interoceptive and exteroceptive inputs [11,62]. Heydrich et al. (2021) found that, under the synchronized mind-vision conditions, individuals with higher empathy ability were able to swiftly switch perspectives between the virtual bodies of self and those of others, which relied heavily on the capacity to perceive others’ situations. Taken together, the available evidence, deriving primarily from the research of empathy and self-processing, suggests that the interaction between basic forms of self-processing (interoceptive and exteroceptive processing) and higher-level self-processing (mental self-processing) illuminates the distinct representations of BSC across individuals.

These findings reveal that processes of self-processing do not operate in isolation but rather interact dynamically to give rise to a unified yet varied sense of BSC across individuals in a multilevel model. The understanding of individual differences in BSC can be enriched by considering the complex interplay of various self-processing mechanisms that involve multiple levels.

## 4. Neural Basis of Embodied Individual Differences in BSC

Notably, convergent evidence suggests that variations in brain structure in terms of morphology and degree of brain functional connectivity and task-based activation may underlie individual differences in BSC [129,134]. In healthy populations, the processing of BSC shares similar neural patterns and brain mechanisms. On the one hand, as for the brain structures (morphology), individual differences in BSC can be explained by differences in cortical thickness [134,135]. Specifically, the subject’s intensity of body ownership is positively correlated with the thickness of the areas like left posterior insula, left lateral occipital cortex, bilateral MTG, bilateral posterior central gyrus, left medial orbitofrontal cortex, and bilateral precuneus [77,126,134]. Among these, the insula, postcentral gyrus, and precuneus cortex have also been shown to be implicated in the sense of agency and other complex cognitive functions [135,136,137]. On the other hand, static functional connectivity and task-based activation provide comprehensive views of brain function, capturing both the immediate responses to external stimuli and the enduring framework that supports cognition and behavior [21,138]. For example, studies have shown that impaired connectivity within the angular gyrus (AG) and TPJ results in a reduced capacity for individuals to localize their body [139]. In addition, during an audio–tactile interaction task, individuals with more restricted peripersonal space exhibited increased functional connectivity between the premotor cortex (PMC) and the left inferior parietal lobe, as well as between the PMC and the right frontal eye fields, when processing stimuli in near space [140].

Based on relevant studies of brain structure (morphology), functional connectivity, and task-based activation, we will subsequently provide an examination of the neural basis of individual differences in the three levels of self-processing, as well as the interaction among various levels (Table 2). We aim to dissect the multifaceted nature of this cognitive mechanism by exploring the various brain regions that are implicated. We are committed to conducting an in-depth examination at various levels, with the intention of identifying the precise neural basis that underlies individual differences in BSC. This will involve searching the specific areas of the brain, their activities, and how these relate to personal behavior and the mind, thereby providing valuable insights into the fundamental nature of human bodies, features, and consciousness (Table 3).

### 4.1. Interoceptive Processing

With respect to variations in interoceptive sensibility, apart from the role of insula, recent evidence has revealed the general negative correlation between interoceptive sensitivity and the functional connectivity in the visual areas, especially in the frontal lobe and basal ganglia (BG) [31]. Moreover, studies have put forward the perspective that the decline in bodily representation ability among individuals with heightened interoceptive sensitivity as they age could be attributed to changes in functional brain networks, such as diminished integrity of the visuospatial network [141,142]. Regarding interoceptive accuracy, as mentioned earlier, the level of interoceptive accuracy is positively correlated with the connectivity of the ACC or insula [30]. In addition, Wang et al. (2019) discovered that the task-based activation of the anterior insula cortex was predictive of individual differences in interoceptive accuracy. When it comes to interoceptive awareness, research has identified a strong correlation between interoceptive awareness and regional gray matter volume in the operculum and right anterior insula, which mediate the explicit awareness towards internal bodily sensations [143,144,145].

These results underscore the complex brain mechanisms underlying interoception and suggest that individual differences in interoceptive dimensions may have significant implications for cognitive experiences, as well as bodily self-perception.

### 4.2. Exteroceptive Processing

Keenan et al. (2000) discovered that the right PFC is preferentially involved in self-face recognition [146]. Similarly, Morita et al. (2017) found that visual self-face recognition activated the right inferior frontoparietal cortices [147]. It is noteworthy that the insula is pivotal for detecting external sensory stimuli, while the ACC holds a crucial role in modulating responses across sensory, association, and motor cortices [148]. It is crucial to highlight that the insula and anterior cingulate cortex form the integral components of the salience network. This network possesses the ability to process multisensory stimuli and promote task-based information processing, thereby enabling it to engage with and modulate other core networks, such as the attention and cognitive control networks [148,149]. In addition, within the frontal cortex, a particularly robust correlation was observed between the bilateral caudal regions of the ACC and the width of the temporal binding window [33]. Additional evidence implied that the ACC possesses the capacity to integrate information [162]. While still conjectural, this finding may suggest that variations in the temporal binding window could reflect differences in the processes facilitated by the ACC, which are integrated in responding to and selecting among multiple sensory inputs [33]. Moreover, the parietal lobe serves as an integration hub for information derived from primary sensory modalities, thereby facilitating the coordination of appropriate behavioral responses [150]. There was a strong correlation between multisensory integration and the parietal lobe [163], whose anatomical connectivity modulates individual differences in multisensory processing. Furthermore, Guterstam et al. (2015) observed that the activity pattern in the hippocampus reflected the multisensory representation of the bodily self in space [34]. Katayama et al. (2024) found that both the right rostrolateral PFC (rlPFC) and hippocampus displayed increased activation when performing inference tasks related to Bayesian modeling [151].

Overall, these results emphasize the importance of specific brain regions, such as the insula, ACC, parietal lobe, hippocampus, and rlPFC, in processing and integrating sensory information, and how individual differences in these processes may affect cognitive and behavioral outcomes in the context of BSC.

### 4.3. Mental Self-Processing

The maintenance and development of individuals’ self-consciousness and self-identity identification are dependent on the regulation of specific brain regions, including the dorsomedial PFC (dmPFC), MTG, inferior frontal gyrus (IFG), and AG [35]. These regions are involved in semantic processing, combining the self and past life experience [153,154]. Additionally, research has observed that the ventromedial PFC (vmPFC) is integrated in processing self-relevant information and maintaining self-concept coherence [155]. The vmPFC plays a crucial role in social cognitive processes involving self and others [109]. It is part of the default mode network (DMN), which is commonly activated during the execution of cognitive tasks [148,149]. Regarding the individual differences in sociocultural construction, research has identified the neural basis that differentiates between independent and interdependent self-construal concepts [32,152]. Increased neural activities were found within the dmPFC, lateral PFC (LPFC), and TPJ among East Asians, who are typically associated with interdependent self-construal. In contrast, greater activation was observed in the ACC, vmPFC, bilateral insula, and right temporal pole among Westerners, who tend to embody independent self-construal. Moreover, the precuneus, a key component of the DMN, plays a crucial role in linking the BSC to socially related emotional information, as well as mediating self-referential processing [157].

These findings highlight the role of specific brain regions in self-consciousness and identity, as well as cultural differences in the neural representations of self-concepts.

### 4.4. Interaction of Multiple Levels of Self-Processing

In the integrated body system, interoception and exteroception are harmoniously blended [1]. During a cardio–visual feedback task, correlation analysis showed that the intraparietal sulcus (IPS) (a region nestled within the parietal lobe) and PMC delved deep into the interaction between interoception and exteroception [1]. Their joint contribution forms a neural scaffolding upon which the multifaceted phenomenon of BSC is constructed. Additionally, previous studies suggested that the insula and ACC play a crucial role in complex social emotions, such as empathy [158]. Specifically, activation in the insula was positively correlated with individuals’ ratings regarding their empathy ability, which offers a more profound and nuanced perspective when comprehending the substantial influence that variances in empathy ability exert on the manner in which individuals process interoception and exteroception [159]. Moreover, Seitz et al. (2006) observed that the awareness and comprehending of the intentions of others was shown to involve the rostral PFC [160]. Lavarco et al. (2022) found that the right TPJ is the specific region responsible for facilitating the comprehension of self-conscious emotions such as empathy [161].

These insights emphasize the connection between interoception, exteroception, and social emotions, suggesting that the brain regions involved in these processes are critical for both unique BSC and distinctive emotional experiences.

## 5. Future Directions

Our review comprehensively contemplated individual differences in BSC, guided by a three-level-self model presented above that allowed representing the relationship between various bodily experiences and self-processing. However, some issues should be noted. Firstly, despite longstanding philosophical and scientific interest in various aspects of bodily experiences, the variations in agency due to individual differences in the context of BSC are not yet fully explored and discussed. Secondly, at odds with studies on brain structure in terms of morphology and resting-state functional connectivity, we noted a lack of studies into task-based brain activation in building individual differences in BSC. Hence, these limitations open new research directions for future work, which is suggested to systematically consider the individual differences in thorough aspects of BSC and to explore how different brain patterns influence the different experience of the bodily self, as these experimental studies are still rather scarce. Thirdly, it has been observed that, using the RHI paradigm, the majority of studies on empathy have primarily involved Black and White participants, potentially limiting the generalizability of findings. This limitation arises because differences in the results could be attributed to the contrast between the skin color of the rubber hand and that of the participants’ own hand, rather than being due to semantic associations with skin color. To enhance the reliability of results, future studies should consider introducing an additional skin tone to broaden the diversity of the sample and reinforce the validity of the existing findings. Moreover, our review explores the enhancement, formation, and modification of BSC, with minimal discussion of its potential suppression or destruction. Notably, practices such as meditation or ingestion of hallucinogens can significantly impair BSC, leading to a condition of disoriented consciousness [164,165]. These different states of BSC show that its essence is highly plastic, and some diseases or drugs can be studied by intervening or changing BSC. Moreover, we are supposed to explore both typical and atypical manifestations of BSC, allowing for more precise investigations into BSC among specific populations. Specifically, as regards other possible future directions, we are convinced that forthcoming studies should try to focus on clinical patients, such as those afflicted with depersonalization disorder (DPD), who struggle to experience genuine sensation inside their bodies. Interventions designed to enhance bodily perception (e.g., mindfulness) could be utilized to aid individuals in regulating their emotions and fostering a sense of mind–body integration [166,167]. In addition, supernumerary phantom limbs is a special form of BSC, which refers to the experience of fantasizing about having extra body parts. In this case, it is often difficult for patients to identify the real physical limbs [168]. Similarly, autotopagnosia, an inability to discriminate and localize parts of the body, reflects the failure to integrate information from one side of the body [169]. We believe that BSC research is helpful to understand the mechanism of these diseases or special states.

As discussed above, this review provides informational value for future research in the clinical field. Intriguingly, the research of BSC from the perspective of individual differences can offer innovative viewpoints and insights into particular diseases marked by disturbances in self-perception, thereby providing more enhanced comprehension and potential therapeutic strategies [170,171]. For example, compared to those with Autism-Spectrum Disorder (ASD), healthy subjects possess narrower temporal binding windows, exhibiting heightened sensitivity to sensory signals, which facilitates the superior integration of multisensory information [172,173]. Accordingly, examining the threshold of the temporal binding window, which refers to the boundary where typical individuals transition towards those with ASD, promises to be a fruitful avenue for future investigative pursuits. Together, the academic community should endeavor to delve deeper into the underlying mechanisms involved in the transition from healthy subjects to patients. It might be worth contemplating the integration of extensively employed robotics and virtual reality technologies with clinical treatments, brain imaging methods, and cognitive neuroscience [174,175], in order to amalgamate diverse methodologies of significant practical relevance.

## 6. Conclusions

Pivotally, we summarized the factors related to individual differences in BSC through the conceptual framework of the three-level-self model, in line with the perspectives of single-level processing and the interaction of multilevel processing. These include the perceptibility of internal bodily signals, the processing principles of multisensory signals, the personal traits shaped by cultural and environmental factors, and the interaction modes among multiple levels of self-processing. Furthermore, research has highlighted the significance of the insula and ACC in multiple levels of self-processing, engaging in body sensation formation and emotional comprehension. Moreover, in interoceptive processing, changes in regions like the operculum are strongly correlated with variations in interoception. As for exteroceptive processing, the parietal lobe, rlPFC, and hippocampus play a crucial role in coordinating sensory signals and Bayesian modeling. Mental self-processing relies on regions like the MPFC. Interoceptive and exteroceptive interactions vary among individuals, occurring within regions like the PMC. In sum, we conclude that the presence of differences in individual characteristics moderates the perception of bodily being, thereby explaining diverse experiences of BSC. Following this perspective, the use of advanced technologies is put forward to facilitate early diagnosis and screening of various clinical disorders, enabling timely interventions to improve patient outcomes.

## Figures and Tables

**Figure 1 brainsci-14-00795-f001:**
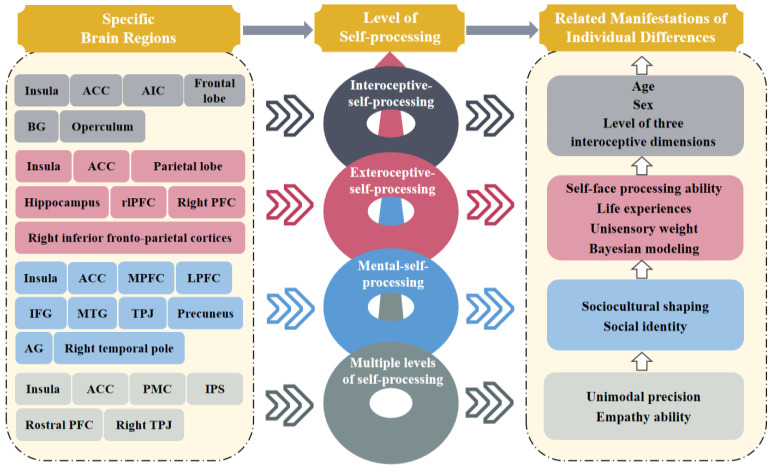
Specific brain regions among various levels of self-processing and their related manifestations of individual differences. Note: ACC: anterior cingulate cortex, AIC: anterior insular cortex, BG: basal ganglia, rlPFC: rostrolateral prefrontal cortex, MPFC: medial prefrontal cortex, LPFC: lateral prefrontal cortex, IFG: inferior frontal gyrus, MTG: middle temporal gyrus, TPJ: temporoparietal junction, AG: angular gyrus, PMC: premotor cortex, IPS: intraparietal sulcus.

**Table 1 brainsci-14-00795-t001:** The manifestations of individual differences and related varied BSC.

Manifestations of Individual Differences		Related Individual Differences in BSC
Internal bodily signal perceptibility	Interoceptive accuracy	Higher interoceptive accuracy predicted narrower peripersonal space [39]. Higher interoceptive accuracy indirectly correlated with stronger intensity of body illusions [40,41]. When exposed to bodily self-stimuli, individuals with higher interoceptive accuracy may facilitate body identification and ownership [42].
	Interoceptive sensibility	Levels of interoceptive sensibility affected the vividness of the RHI [43]. Interoceptive sensibility produced negative effects in body representation as individuals aged [44]. Females had higher interoceptive sensibility linked to larger insula volume, which contributes to various aspects of BSC [45].
	Interoceptive awareness	Males had greater interoceptive awareness, making them less prone to body illusions than females [46].
Multisensory processing principles	Bayesian CI principles (unisensory signal weighting)	Individuals reliant on proprioception demonstrated enhanced body awareness and superior posture perception [15]. Individuals who assign higher weights for proprioceptive signals have a heightened sense of body part localization and are less prone to body illusions [47,48].
	Bayesian CI principles (multisensory processing)	Bayesian CI computational principles explained the phenomenon that individuals with wider temporal binding windows experience a stronger RHI [49,50,51].
	Distinction mechanism of self-other	Individuals with higher AQ were slower to recognize facial expressions, but the reaction time to facial expressions under the self-face condition is faster than that under the non-self-face condition [52,53].
Personal traits shaped by environment	Personality affected by sociocultural constructs	East Asians are inclined to link the observed facial features with the social concept of “face” [16]. Japanese deeply ingrain formality and normativity into their traits, thereby overlooking their actual bodily sensations [54]. Americans tend to closely monitor their physical sensations because of their culture championing freedom and independence [55].
	Personal social identity	White participants experienced more difficulty and took longer to mentally incorporate the image of a black rubber hand into their body schema [56]. Women may be more prone to develop negative self-identification regarding their bodies [57].
Interaction modes	Interaction between low-level self-processing	Individuals with higher interoceptive accuracy showed less decay in their prediction errors and lower exteroceptive precision, leading to reduced vividness in the RHI and whole-body illusion [58].
	Interaction between low and high-level self-processing	Higher empathy correlates with superior bodily signal interpretation, aiding BSC evaluation [59,60]. Individuals with higher empathy ability exhibited a more pronounced experience of the RHI [61]. Individuals with higher empathy ability experienced stronger self-localization drift and illusory self-identification [60]. Individuals with higher empathy can switch perspectives between virtual bodies under synchronized mind-vision conditions [62].

Note: BSC: bodily self-consciousness, RHI: rubber hand illusion, Bayesian CI: Bayesian causal inference, AQ: autism-spectrum quotients.

**Table 2 brainsci-14-00795-t002:** Level of self-processing and its corresponding neural basis and brain structure and function.

Level of Self-Processing	Corresponding Neural Basis	Brain Structure and Function
Interoceptive- processing	Level of interoceptive accuracy is indirectly linked to BSC via ACC and insula connectivity [30,82]. Wang et al. (2019) found that task-based activation of the AIC predicted variations in interoceptive accuracy [88]. A negative correlation was found between interoceptive sensitivity and functional connectivity in the frontal lobe and BG [31]. The decline in bodily representation ability with age among those with higher interoceptive sensitivity may be due to altered brain network function like in the visuospatial network [141,142]. A strong correlation existed between interoceptive awareness and regional gray matter volume in the operculum and right anterior insula [143,144,145].	Functional connectivity in the ACC and insula Task-based activation of the AIC Functional connectivity in the frontal lobe and BG Functional connectivity in the visuospatial network Gray matter volume in the operculum and right anterior insula
Exteroceptive- processing	Keenan et al. (2000) and Morita et al. (2017) suggested that the right PFC and right inferior frontoparietal cortices are involved in self-face recognition [146,147]. Both the insula and ACC are essential for filtering and processing sensory information [148,149]. There is a strong correlation between the bilateral caudal regions of the ACC within the frontal cortex and the width of the temporal binding window [33]. The parietal lobe integrates sensory information, aiding in coordinating behavioral responses [150]. The salience network can process multisensory stimuli and modulate other core networks like the attention and cognitive control networks [148,149]. Activity in the hippocampus reflected the multisensory representation of the bodily self in space [34]. Katayama et al. (2024) suggested that the rlPFC and hippocampus showed increased activation for Bayesian modeling inference tasks [151].	Task-based activation of the right PFC and right inferior frontoparietal cortices Functional connectivity in the insula and ACC Task-based activation of the bilateral caudal regions of the ACC Task-based activation of the parietal lobe Task-based activation of the salience, attention, and cognitive control networks Task-based activation of the rlPFC and hippocampus
Mental-self-processing	East Asians typically show increased neural activity in the dmPFC, LPFC, and TPJ associated with their interdependent self-construal, while Westerners often exhibit greater activation in the ACC, vmPFC, bilateral insula, and right temporal pole related to their independent self-construal [32,152]. Self-consciousness and self-identity recognition rely on specific brain regions for regulation, including dmPFC, MTG, IFG, and AG, which are involved in processing semantics and integrating personal identity with past experiences [35,153,154]. The vmPFC, part of the DMN, plays a role in processing self-relevant information and social cognitive tasks, as well as maintaining coherence of the self-concept [109,148,155]. Lanius et al. (2015) and Cabanis et al. (2013) suggested that the precuneus, a key component of the DMN, is vital for linking the BSC to socially related emotional information, as well as mediating self-referential processing [156,157].	Task-based activation of the dmPFC, LPFC, and TPJ Task-based activation of the ACC, vmPFC, bilateral insula, and right temporal pole Task-based activation of the dmPFC, MTG, IFG, and AG Task-based activation of the vmPFC Task-based activation of the precuneus
Multiple levels of self-processing	The integrated body system harmoniously blends interoception and exteroception, with the IPS and PMC contributing to this interaction [1]. The insula and ACC play a crucial role in complex social emotions such as empathy, with activation in the insula positively correlated with empathy ability [158,159]. Seitz et al. (2006) suggested that the awareness and comprehension of the intentions of others was shown to involve the rostral PFC [160]. Lavarco et al. (2022) suggested that the right TPJ is responsible for comprehension of self-conscious emotions like empathy [161].	Task-based activation of the IPS and PMC Task-based activation of the insula and ACC Task-based activation of the PFC Task-based activation of the TPJ

Note: BSC: bodily self-consciousness, ACC: anterior cingulate cortex, AIC: anterior insular cortex, BG: basal ganglia, rlPFC: rostrolateral prefrontal cortex, dmPFC: dorsomedial prefrontal cortex, LPFC: lateral prefrontal cortex, TPJ: temporoparietal junction, vmPFC: ventromedial prefrontal cortex, MTG: middle temporal gyrus, IFG: inferior frontal gyrus, AG: angular gyrus, DMN: default mode network, IPS: intraparietal sulcus, PMC: premotor cortex.

**Table 3 brainsci-14-00795-t003:** Correlation between level of self-processing, brain areas, brain activities, personal behavior and mind, bodily experiences, and features of consciousness.

Level of Self-Processing	Specific Areas of the Brain	Brain Network	Brain Activities	Personal Behavior and Mind	Bodily Experiences	Features of Consciousness
Interoceptive- processing	Insula (for the young)/ACC (for the elderly)	Salience network	Functional connectivity	Perception of heartbeat	Subjective perception of bodily awareness	Context- dependent (for the young)/Context- independent (for the elderly)
	AIC	Salience network	Task-based activation	Performance accuracy on the interoceptive task	Interoceptive accuracy to bodily signals	Perceptibility and clarity of consciousness
	Frontal lobe/BG	Default mode network/(/)	Functional connectivity	Self-reported sensitivity to interoceptive information	Sensitivity to internal bodily signals	Sensibility of consciousness
	Visuospatial network	Visuospatial network	Functional connectivity	Maintaining normal cognitive function	Bodily representation ability	Degree of integration and association
	Operculum/right anterior insula	Sensorimotor/salience network	Gray matter volume	Attention to either heart-beat timing or external note quality	Awareness of bodily and stress responses	Accuracy of interoceptive awareness
Exteroceptive- processing	Right PFC/right inferior frontoparietal cortices	Frontoparietal network	Task-based activation	Self-face recognition and facial expression processing	Sense of identification and ownership with one’s own body	Stable, self-bias, and continually updated
	Insula/ACC	Salience network	Functional connectivity	Attention and cognitive control	Focus on sensory signals and bodily state	Integration of bottom-up and top-down sensory information
	ACC (bilateral caudal regions)	Salience network	Task-based activation	Integrating information over long timescales	Fine resolution of multisensory signals	Efficiency of multisensory processing
	Parietal lobe	Default mode network	Task-based activation	Receiving input from sensory-specific cortices	Perception of multisensory input	Representation of multisensory modalities
	Hippocampus	/	Task-based activation	Multisensory representation of the bodily self in space	Bodily self-location and out-of-body experience	Updatable and plastic
	rlPFC/hippocampus	Executive Control Network/(/)	Task-based activation	Performing inference tasks related to Bayesian modeling	/	Hierarchical and complex
Mental-self- processing	dmPFC/LPFC/TPJ	Default mode/frontoparietal network/(/)	Task-based activation	Self-relevance encoding during social affective process	Sensitive to bodily information related to significant others	Context- dependent
	ACC/vmPFC/bilateral insula/right temporal pole	Salience/default mode/(/)/(/)	Task-based activation	Self-relevance encoding during social affective process	Attention to self-focused bodily information	Context- independent
	dmPFC/MTG/IFG/AG	Default mode/auditory/language/visual recognition network	Task-based activation	Processing semantics and integrating personal identity	Sense of identity and distinction of bodily parts	Coherence and stability of self- consciousness
	vmPFC	Default mode network	Task-based activation	Caring about self-evaluation	Sensitive to bodily traits related to self-identity	Constructed by social identity
Multiple levels of self-processing	Insula/ACC	Salience network	Task-based activation	Experiences of complex social emotions	Compassion for physical pain	High-level cognitive processing
	PMC/IPS	Motor/frontoparietal network	Task-based activation	Preferentially processing self-related signals	Self-location and identification	Global unity and temporal continuity
	Rostral PFC	Default mode network	Task-based activation	Comprehending others’ intentions	/	Emotion- dependent
	Right TPJ	/	Task-based activation	Comprehension of self-conscious emotions	/	Impacted by emotions

Note: ACC: anterior cingulate cortex, AIC: anterior insular cortex, BG: basal ganglia, PFC: prefrontal cortex, rlPFC: rostrolateral prefrontal cortex, dmPFC: dorsomedial prefrontal cortex, LPFC: lateral prefrontal cortex, TPJ: temporoparietal junction, vmPFC: ventromedial prefrontal cortex, MTG: middle temporal gyrus, IFG: inferior frontal gyrus, AG: angular gyrus, PMC: premotor cortex, IPS: intraparietal sulcus.

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
