# Peer review of "Individual Differences in Bodily Self-Consciousness and Its Neural Basis"

_brainsci, 2024, doi:10.3390/brainsci14080795_

Round 1

Reviewer 1 Report

Comments and Suggestions for Authors

This is an interesting review, but is quite thin in both the material covered and the perspective taken.  RBI and body illusion studies are interesting, but are not the only of investigating BSC.  The paper would be improved by a broader perspective.

p. 3, Fig. 1:  It is surprising to see no mention of rostral PFC (BA 10) or of hippocampus anywhere in this Figure, especially with regard to Bayesian modeling and source monitoring or ownership detection.

p. 9-15, Table 1:  The text in this table duplicates the text in the surrounding sections.  The table is very long and seems unneccessary.

There are many references to individual brain areas, but no mention of how these participate in established functional networks, e.g. the salience, attention, executive, and default networks.  It would be helpful to provide some of this context, e.g. in a table, figure, or both.

There is no discussion of what is going on in clinically-relevant body agnosias, phantom limb syndromes, or even local anesthesia.  Considering these conditions would add significantly to the paper.

There is, similarly, no discussion of enhanced, inhibited, or modified BSC in meditative states, with halluncinogens, or in conditions such as out-of-body experiences.  Some mention of these, with approprioate references, would also improve the paper.

Finally, some mention should be made of how BSC interfaces to other forms of self-consciousness, e.g. autobiographical narrative or the sense of identity through time.

Comments on the Quality of English Language

Some grammatical errors need fixing.

Author Response

Authors’ Response to Reviewer Comments

We are truly grateful to the editors for coordinating the review of our manuscript and to the reviewers for their insightful and constructive comments. By making thorough and careful revisions to the manuscript based on the reviewers’ comments, we believe that this manuscript has been greatly improved.

We have made several improvements to the manuscript: (1) The table mentioned by the reviewers was carefully revised to enhance pertinence. We added three tables about relationship between self-processing and BSC, its neural basis and the specific manifestations of individual differences. (2) We have revised the whole statement to avoid conceptual ambiguity. The sentences mentioned by the reviewer (e.g., the concept of self) were carefully revised to enhance clarity.

Our point-by-point responses to the comments of the reviewers are listed below. The changes in the revised manuscript are marked in blue.

Reviewer #1:

This is an interesting review, but is quite thin in both the material covered and the perspective taken. RBI and body illusion studies are interesting, but are not the only of investigating BSC. The paper would be improved by a broader perspective.

Thanks for the reviewer’s patience and pointing out the problems, which are needed to be fixed up further and make sure the correction of the expression.

Comment 1: p. 3, Fig. 1: It is surprising to see no mention of rostral PFC (BA 10) or of hippocampus anywhere in this Figure, especially with regard to Bayesian modeling and source monitoring or ownership detection.

Thank you for this comment. Following your suggestion, we have revised the paper about how self-processing involves rostral PFC and hippocampus. Please see page 17, lines 517–521:

“Furthermore, Guterstam et al. (2015) observed that the activity pattern in hippocampus reflected the multisensory representation of the bodily self in space. Katayama et al. (2024) found that both the right rostrolateral PFC (rlPFC) and hippocampus displayed increased activation when performing inference tasks related to Bayesian modeling.”

Please see page 18, lines 558-560.

“Seitz et al. (2006) observed that the awareness and comprehending of the intentions of others was shown to involve rostral PFC.”

Additionally, We have added information of rostral PFC and hippocampus in Figure 1.

(The picture can't be displayed here, please see the attachment for details.)

Figure 1. Specific Brain Regions among Various Levels of Self-processing and Their Related Manifestations of Individual Differences. Note: ACC: anterior cingulate cortex, AIC: anterior insular cortex, BG: Basal ganglia, rlPFC: rostrolateral prefrontal cortex, dmPFC: dorsomedial prefrontal cortex, LPFC: lateral prefrontal cortex, TPJ: temporoparietal junction, vmPFC: ventromedial prefrontal cortex, MTG: middle temporal gyrus, IFG: inferior frontal gyrus, AG: angular gyrus, IPS: intraparietal sulcus, PMC: premotor cortex

Comment 2: p. 9-15, Table 1: The text in this table duplicates the text in the surrounding sections. The table is very long and seems unneccessary.

Thank you for this comment. We acknowledged that the Table 1 was very long and seemed to just repeat info from text verbatim. Thus, considering the opinions of the other reviewer, we have revised the table according to your suggestions, and showed the necessary information such as the Manifestations of Individual Differences and Related Varied BSC in three tables (Table 1), Corresponding Neural Basis of Self-processing (Table 2), the Features of Brain Regions and BSC to Each Self-processing Level (Table 3).

Table 1. The Manifestations of Individual Differences and Related Varied BSC

Manifestations of Individual Differences

Related Individual Differences in BSC

Internal bodily signal perceptibility

Interoceptive accuracy

·Higher interoceptive accuracy predicted narrower peripersonal space (Ardizzi & Ferri, 2018).

·Higher interoceptive accuracy indirectly correlated with stronger intensity of body illusions (Mai et al., 2018; Park et al., 2016).

Interoceptive sensibility

· Levels of interoceptive sensibility affected the vividness of RHI (Tsakiris, 2011).

· Interoceptive sensibility produced negative effects in body representation as individuals aged (Raimo, 2021).

· Females had higher interoceptive sensibility linked to larger insula volume, which contributes to various aspects of BSC (Longarzo et al., 2021).

Interoceptive awareness

· Males had greater interoceptive awareness, making them less prone to body illusions than females (Palmer & Tsakiris, 2018).

Multisensory processing principles

Bayesian causal inference principles (unisensory signal weighting)

·Individuals reliant on proprioception demonstrated enhanced body awareness and superior posture perception (Virtanen et al., 2022; Ehrsson, 2012).

· Individuals who assign higher weights for proprioceptive signals have a heightened sense of body part localization and are less prone to body illusions (Botan et al., 2021; Ernst & Banks, 2002).

Bayesian causal inference principles (multisensory processing)

·Bayesian CI computational principles explained the phenomenon that individuals with wider temporal binding windows experience stronger RHI (Litwin, 2020; Chancel et al., 2021, 2022).

Personal traits shaped by environment

Personality affected by sociocultural constructs

· East Asians are inclined to link the observed facial features with the social concept of “face” (Maister & Tsakiris, 2014).

· Japanese deeply ingrain formality and normativity into their traits, thereby overlooking their actual bodily sensations (Freedman et al., 2020).

·Americans tend to closely monitor their physical sensations because of their culture championing freedom and independence (Freedman et al., 2020).

Personal social identity

· White participants experienced more difficulty and took longer to mentally incorporate the image of a black rubber hand into their body schema (Lira et al., 2017).

· Women may be more prone to develop negative self-identification regarding their bodies (Kling et al., 2018).

Interaction modes

Interaction between low-level modals

·Individuals with higher interoceptive accuracy showed less decay in their prediction errors and lower exteroceptive precision, leading to reduced vividness in the RHI and whole-body illusion (Ainley et al., 2016).

Interaction between low and high-level modals

· Higher empathy correlates with superior bodily signal interpretation, aiding BSC evaluation (Mul et al., 2018; Fukushima et al., 2011).

· Individuals with higher empathy ability exhibited a more pronounced experience of RHI (Asai et al., 2011).

· Individuals with higher empathy ability experienced stronger self-localization drift and illusory self-identification (Mul et al., 2019).

· Individuals with higher empathy can switch perspectives between virtual bodies under synchronized mind-vision conditions (Heydrich et al., 2021).

Note: BSC: bodily self-consciousness, RHI: rubber hand illusion, Bayesian CI: Bayesian causal inference

Table 2. Level of Self-processing and its Corresponding Neural Basis and Brain Structure and Function

Level of Self-processing

Corresponding Neural Basis

Brain Structure and Function

Interoceptive-

processing

· Level of interoceptive accuracy is indirectly linked to BSC via ACC and insula connectivity (Ueno et al., 2020; Chong et al., 2017).

· Wang et al. (2019) found that task-based activation of the AIC predicted variations in interoceptive accuracy.

· A negative correlation was found between interoceptive sensitivity and functional connectivity in frontal lobe and BG (Smith et al., 2022).

· The decline in bodily representation ability with age among those with higher interoceptive sensitivity may be due to altered brain network function like visuospatial network (Betzel et al., 2014; Bagarinao et al., 2019).

· A strong correlation existed between interoceptive awareness and regional gray matter volume in operculum and right anterior insula (Pollatos et al., 2007; Critchley et al., 2004; Caseras et al., 2013).

· Functional connectivity in ACC and insula

· Task-based activation of AIC

· Functional connectivity in frontal lobe and BG

· Functional connectivity in visuospatial network

· Gray matter volume in operculum and right anterior insula

Exteroceptive-

processing

· Both insula and ACC are essential for filtering and processing sensory information (Menon & Uddin, 2010; Yantis, 2008).

· There is a strong correlation between the bilateral caudal regions of the ACC within the frontal cortex and the width of the temporal binding window (Johnston et al., 2022).

· Parietal lobe integrates sensory information, aiding in coordinating behavioral responses (Driver & Noesselt, 2008).

· The salience network can process multisensory stimuli and modulate other core networks like the attention and cognitive control networks (Menon & Uddin, 2010; Yantis, 2008).

· Activity in hippocampus reflected the multisensory representation of the bodily self in space (Guterstam et al., 2015).

· rlPFC and hippocampus showed increased activation for Bayesian modeling inference tasks (Katayama et al., 2024).

· Functional connectivity in insula and ACC

· Task-based activation of the bilateral caudal regions of the ACC

· Task-based activation of parietal lobe

· Task-based activation of salience, attention and cognitive control networks

· Task-based activation of rlPFC and hippocampus

Mental-self-

processing

· East Asians typically show increased neural activity in the dmPFC, LPFC, and TPJ associated with their interdependent self-construal, while Westerners often exhibit greater activation in the ACC, vmPFC, bilateral insula, and right temporal pole related to their independent self-construal (Han & Ma, 2014; Chiao et al., 2009).

· Self-consciousness and self-identity recognition rely on specific brain regions for regulation, including dmPFC, MTG, IFG, and AG, which involved in processing semantics and integrating personal identity with past experiences (D'Argembeau et al., 2014; Binder et al., 2009; Price, 2010).

· The vmPFC, part of the DMN, plays a role in processing self-relevant information and social cognitive tasks, as well as maintaining coherence of self-concept (Elder et al., 2023; Amodio & Frith, 2006; Menon & Uddin, 2010).

· The precuneus, a key component of DMN, is vital for linking the BSC to socially-related emotional information, as well as mediating self-referential processing (Lanius et al., 2015; Cabanis et al., 2013).

· Task-based activation of dmPFC, LPFC, and TPJ

· Task-based activation of ACC, vmPFC, bilateral insula, and right temporal pole

· Task-based activation of dmPFC, MTG, IFG, and AG

· Task-based activation of vmPFC

· Task-based activation of precuneus

Multiple levels of self-processing

· The integrated body system harmoniously blends interoception and exteroception, with the IPS and PMC contributing to this interaction (Park & Blanke, 2019).

· The insula and ACC play a crucial role in complex social emotions such as empathy, with activation in insula positively correlated with empathy ability (Immordino-Yang et al., 2009; Singer et al., 2004).

· The awareness and comprehending of the intentions of others was shown to involve rostral PFC (Seitz et al., 2006).

· The right TPJ is responsible for comprehension of self-conscious emotions like empathy (Lavarco et al., 2022).

· Task-based activation of IPS and PMC

· Task-based activation of insula and ACC

· Task-based activation of PFC

· Task-based activation of TPJ

Note: ACC: anterior cingulate cortex, AIC: anterior insular cortex, BG: Basal ganglia, rlPFC: rostrolateral prefrontal cortex, dmPFC: dorsomedial prefrontal cortex, LPFC: lateral prefrontal cortex, TPJ: temporoparietal junction, vmPFC: ventromedial prefrontal cortex, MTG: middle temporal gyrus, IFG: inferior frontal gyrus, AG: angular gyrus, DMN: default mode network, IPS: intraparietal sulcus, PMC: premotor cortex.

Table 3. Correlation Between Level of Self-Processing, Brain Areas, Brain networks, Brain Activities, Personal Behavior and Mind, Bodily Experiences, and Features of Consciousness

Level of self-processing

Specific areas of the brain

Brain network

Brain activities

Personal behavior and mind

Bodily experiences

Features of consciousness

Interoceptive-

processing

Insula (for the young) / ACC (for the elderly)

Salience network

Functional connectivity

Perception of heartbeat

Subjective perception of bodily awareness

Context-

dependent (for the young) / Context-

independent (for the elderly)

AIC

Salience network

Task-based activation

Performance accuracy on the interoceptive task

Interoceptive accuracy to bodily signals

Perceptibility and clarity of consciousness

Frontal lobe / BG

Default mode network / (/)

Functional connectivity

Self-reported sensitivity to interoceptive information

Sensitivity to internal bodily signals

Sensibility of consciousness

Visuospatial network

Visuospatial network

Functional connectivity

Maintaining normal cognitive function

Bodily representation ability

Degree of integration and association

Operculum / right anterior insula

Sensorimotor / salience network

Gray matter volume

Attention to either heart-beat timing or external note quality

Awareness of bodily and stress responses

Accuracy of interoceptive awareness

Exteroceptive-

processing

Insula / ACC

Salience network

Functional connectivity

Attention and cognitive control

Focus on sensory signals and bodily state

Integration of bottom-up and top-down sensory information

ACC

(bilateral caudal regions)

Salience network

Task-based activation

Integrating information over long timescales

Fine resolution of multisensory signals

Efficiency of multisensory processing

Parietal lobe

Default mode network

Task-based activation

Receiving input from sensory-specific cortices

Perception of multisensory input

Representation of multisensory modalities

Hippocampus

/

Task-based activation

Multisensory representation of the bodily self in space

Bodily self-location and out-of-body experience

Updatable and plastic

rlPFC / hippocampus

Executive Control Network / (/)

Task-based activation

Performing inference tasks related to Bayesian modeling

/

Hierarchical and complex

Mental-self-

processing

dmPFC / LPFC / TPJ

Default mode / frontoparietal network / (/)

Task-based activation

Self-relevance encoding during social affective process

Sensitive to bodily information related to significant others

Context-

dependent

ACC / vmPFC/ bilateral insula / right temporal pole

Salience / default mode / (/) / (/)

Task-based activation

Self-relevance encoding during social affective process

Attention to self-focused bodily information

Context-

independent

dmPFC / MTG / IFG / AG

Default mode / auditory / language / visual recognition network

Task-based activation

Processing semantics and integrating personal identity

Sense of identity distinction of bodily parts

Coherence and stability of self-

consciousness

vmPFC

Default mode network

Task-based activation

Caring about self-evaluation

Sensitive to bodily traits related to self-identity

Constructed by social identity

Multiple levels of self-processing

Insula / ACC

Salience network

Task-based activation

Experiences of complex social emotions

Compassion for physical pain

High-level cognitive processing

PMC / IPS

Motor / frontoparietal network

Task-based activation

Preferentially processing self-related signals

Self-location and identification

Global unity  and temporal continuity

Rostral PFC

Default mode network

Task-based activation

Comprehending of the others’ intentions

/

Emotion-

dependent

Right TPJ

/

Task-based activation

Comprehension of self-conscious emotions

/

Impacted by emotions

Note: ACC: anterior cingulate cortex, AIC: anterior insular cortex, BG: Basal ganglia, rlPFC: rostrolateral prefrontal cortex, dmPFC: dorsomedial prefrontal cortex, LPFC: lateral prefrontal cortex, TPJ: temporoparietal junction, vmPFC: ventromedial prefrontal cortex, MTG: middle temporal gyrus, IFG: inferior frontal gyrus, AG: angular gyrus, BSC: bodily self-consciousness, IPS: intraparietal sulcus, PMC: premotor cortex.

Comment 3: There are many references to individual brain areas, but no mention of how these participate in established functional networks, e.g. the salience, attention, executive, and default networks. It would be helpful to provide some of this context, e.g. in a table, figure, or both.

Thank you for this comment. Following your suggestion, we have included a paragraph about how individual brain areas participate in established functional networks, e.g. the salience, attention, cognitive control, and default mode networks. Please see page 17, lines 503–506:

“It is crucial to highlight that the insula and anterior cingulate cortex form the integral components of the salience network. This network possesses the ability to process multisensory stimuli and promote task-based information processing, thereby enabling it to engage with and modulate other core networks, such as the attention and cognitive control networks (Menon & Uddin, 2010; Yantis, 2008).”

Please see page 17, lines 533–535:

“The vmPFC plays a crucial role in social cognitive processes involving self and others (Amodio & Frith, 2006). It is part of the default mode network (DMN), which is commonly activated during the execution of cognitive tasks (Menon & Uddin, 2010).”

Please see page 18, lines 542–544:

“Moreover, the precuneus, a key component of DMN, plays a crucial role in linking the BSC to socially-related emotional information, as well as mediating self-referential processing (Cabanis et al., 2013).”

Additionally, We have added information of these functional networks in Table 3.

Table 3. Correlation Between Level of Self-Processing, Brain Areas, Brain networks, Brain Activities, Personal Behavior and Mind, Bodily Experiences, and Features of Consciousness

Level of self-processing

Specific areas of the brain

Brain network

Brain activities

Personal behavior and mind

Bodily experiences

Features of consciousness

Interoceptive-

processing

Insula (for the young) / ACC (for the elderly)

Salience network

Functional connectivity

Perception of heartbeat

Subjective perception of bodily awareness

Context-

dependent (for the young) / Context-

independent (for the elderly)

AIC

Salience network

Task-based activation

Performance accuracy on the interoceptive task

Interoceptive accuracy to bodily signals

Perceptibility and clarity of consciousness

Frontal lobe / BG

Default mode network / (/)

Functional connectivity

Self-reported sensitivity to interoceptive information

Sensitivity to internal bodily signals

Sensibility of consciousness

Visuospatial network

Visuospatial network

Functional connectivity

Maintaining normal cognitive function

Bodily representation ability

Degree of integration and association

Operculum / right anterior insula

Sensorimotor / salience network

Gray matter volume

Attention to either heart-beat timing or external note quality

Awareness of bodily and stress responses

Accuracy of interoceptive awareness

Exteroceptive-

processing

Insula / ACC

Salience network

Functional connectivity

Attention and cognitive control

Focus on sensory signals and bodily state

Integration of bottom-up and top-down sensory information

ACC

(bilateral caudal regions)

Salience network

Task-based activation

Integrating information over long timescales

Fine resolution of multisensory signals

Efficiency of multisensory processing

Parietal lobe

Default mode network

Task-based activation

Receiving input from sensory-specific cortices

Perception of multisensory input

Representation of multisensory modalities

Hippocampus

/

Task-based activation

Multisensory representation of the bodily self in space

Bodily self-location and out-of-body experience

Updatable and plastic

rlPFC / hippocampus

Executive Control Network / (/)

Task-based activation

Performing inference tasks related to Bayesian modeling

/

Hierarchical and complex

Mental-self-

processing

dmPFC / LPFC / TPJ

Default mode / frontoparietal network / (/)

Task-based activation

Self-relevance encoding during social affective process

Sensitive to bodily information related to significant others

Context-

dependent

ACC / vmPFC/ bilateral insula / right temporal pole

Salience / default mode / (/) / (/)

Task-based activation

Self-relevance encoding during social affective process

Attention to self-focused bodily information

Context-

independent

dmPFC / MTG / IFG / AG

Default mode / auditory / language / visual recognition network

Task-based activation

Processing semantics and integrating personal identity

Sense of identity distinction of bodily parts

Coherence and stability of self-

consciousness

vmPFC

Default mode network

Task-based activation

Caring about self-evaluation

Sensitive to bodily traits related to self-identity

Constructed by social identity

Multiple levels of self-processing

Insula / ACC

Salience network

Task-based activation

Experiences of complex social emotions

Compassion for physical pain

High-level cognitive processing

PMC / IPS

Motor / frontoparietal network

Task-based activation

Preferentially processing self-related signals

Self-location and identification

Global unity  and temporal continuity

Rostral PFC

Default mode network

Task-based activation

Comprehending of the others’ intentions

/

Emotion-

dependent

Right TPJ

/

Task-based activation

Comprehension of self-conscious emotions

/

Impacted by emotions

Note: ACC: anterior cingulate cortex, AIC: anterior insular cortex, BG: Basal ganglia, rlPFC: rostrolateral prefrontal cortex, dmPFC: dorsomedial prefrontal cortex, LPFC: lateral prefrontal cortex, TPJ: temporoparietal junction, vmPFC: ventromedial prefrontal cortex, MTG: middle temporal gyrus, IFG: inferior frontal gyrus, AG: angular gyrus, BSC: bodily self-consciousness, IPS: intraparietal sulcus, PMC: premotor cortex.

Comment 4: There is no discussion of what is going on in clinically-relevant body agnosias, phantom limb syndromes, or even local anesthesia. Considering these conditions would add significantly to the paper.

Thank you for this comment. We have added the clinically-relevant evidence to discussion to add significantly to the paper. Please see page 19, lines 599-606:

“In addition, supernumerary phantom limbs is a special form of BSC, which refers to the experience of fantasizing about having extra body parts. In this case, it is often difficult for patients to identify the real physical limbs (Dieguez & Lopez, 2017). Similarly, the autotopagnosia, an inability to discriminate and localize parts of the body, reflects the failure to integrate information from one side of the body (Basagni et al., 2021). We believe that the BSC research is helpful to understand the mechanism of these diseases or special states.”

Comment 5: There is, similarly, no discussion of enhanced, inhibited, or modified BSC in meditative states, with halluncinogens, or in conditions such as out-of-body experiences. Some mention of these, with approprioate references, would also improve the paper.

Thank you for this comment. Following your suggestion, we have added the evidence of enhanced, modified or impaired BSC in meditative states, with halluncinogens, or in conditions such as out-of-body experiences. Please see page 6, lines 248-252:

“Of course there are some abnormal examples, such as the out-of-body experience caused by the sense of disintegration of multi-sensory stimuli (Aspell et al., 2012). It is a form of consciousness that occurs outside of the physical body and may interfere with some experiences of BSC like self location (Blanke, 2004).”

Please see pages 18-19, lines 586-592:

“Moreover, our review explores the enhancement, formation, and modification of BSC, with minimal discussion of its potential suppression or destruction. Notably, practices such as meditation or ingestion of halluncinogens can significantly impair BSC, leading to a condition of disorienting consciousness (Gutiérrez et al., 2022; Millière et al., 2018). These different states of BSC show that its essence is highly plastic, and some diseases or drugs can be studied by intervening or changing BSC.”

Comment 6: Finally, some mention should be made of how BSC interfaces to other forms of self-consciousness, e.g. autobiographical narrative or the sense of identity through time.

Thank you for this comment. Following your suggestion, we have added the evidence of how BSC interfaces to other forms of self-consciousness, e.g. autobiographical self or diachronic continuity of personal identity. Please see page 7, lines 323-329:

“In addition, BSC was found to be related to the self-consciousness of mental form. Bréchet (2022) found that the autobiographical self connects BSC with situational memory, allowing individuals to re-experience past scenes from the inside or outside of body. Habermas & Köber (2015) suggested that the stability and constancy of the bodily self are conducive to maintaining the coherence of self-awareness, thus supporting the diachronic continuity of personal identity.”

Reviewer 2 Report

Comments and Suggestions for Authors

  • A brief summary 
  • Article provides a neuropsychology analysis of what the authors refer to as "Bodily self-consciousness"  (BSC) concerned with awareness of one’s bodily states. They address the  individual differences in BSC in terms of the underlying factors and neural basis found as accounting for individual differences. They also assert to account for these individual differences in BSC using a three-level-self model, which includes interoceptive, exteroceptive, and mental-self-processing. But they also use this idea of levels of processing to discuss concepts of levels of self and modes of self and to equate the two levels.
  • The authors state: the goal of this review is to establish a comprehensive framework for elucidating the factors associated with self-processing that are linked to the individual differences in BSC, as well as bridging the unique perceptions of BSC with their corresponding neural basis. I do not see that such a comprehensive framework is presented nor the unique perceptions of BCS, or more importantly the different types of forms of BCS-- as opposed to information about variation in activity in neural processing components

Article: highlighting areas of weakness

L33, the very first sentence in the article highlights the overall weakness of the article:

 bodily self-consciousness (BSC) is so rooted in our daily life that it has increasingly become the focus in the fields of 35 philosophy, psychology and neuroscience[1-3].

But the references are just to brain science studies, nothing in philosophy and psychology or other relevant fields that study the self such as neuropsychiatry or consciousness studies.

Authors state that the individual differences of BSC can be traced back to the underlying neural basis of the brain.

I do not see how the article has made a direct presentation of this point, much less how even the different forms of self in the BSC are produced by specific underlying neural bases

Line 56 Individual differences in BSC can be attributed to various aspects.

This presents a claim that I think the authors do not directly address, substantiate, but instead confuscate, mis-represent. Just what are the different types, kind, forms of BSC as opposed to variation in neural processes attributed roles self-related information.

The authors act here as if they are telling is what are these "individual differences" in self but in actuality they are presenting data about individual VARIATION in mechanisms involved in processing information. Variation on some measures of neurological processing, rather than anything that clearly presents what are the major forms of individual difference in self/selves.

Just what is the self, its individual biosocialized variant forms, is not spelled out. Just a lot about what are some of the systems that contribute to the information integrated in the organism that may contribute to self/selves.

The authors refer to a previous study--  we here review previous findings on individual differences in BSC through a three-level-self model--but they do not actually review those previous findings and present them in a systematic way that contributes to this article or furthering the study of the neurological bases of the self and variation in modes of the self.

I have however, read a previous co-authored article by two of the authors (Qin, P.; Wang, M.; Northoff, G. 27 Qin, P.; Wang, M.; Northoff, G. Linking bodily, environmental and mental states in the self-A three-level model based on a meta-analysis. Neurosci Biobehav Rev. 2020, 115, 77–95. ), and I think the current paper requires a brief summary of the core findings from that previous paper in order to provide a neurological model that makes this paper a more comprehensible and a meaningful contribution.

As I elaborate below, I think a presentation (concise) of those previous finding is important for providing the context of what they want to present here.

Authors say: The self as a multimodal entity is embedded in extensive discourses within psychology and philosophy[104,105

[104] Tajadura-Jiménez, A.; Grehl, S.; Tsakiris, M. The other in me: interpersonal multisensory stimulation changes the mental 775 representation of the self. PLoS One. 2012, 7, e40682.

[105] Desbordes, G. Self-related processing in mindfulness-based interventions. Neuron. 2019, 28, 312–316.

These are not considerations within psychology and philosophy. Indeed the paper has ignored a rich amount of information from these fields (psychology and philosophy) as well as psychiatry and psychobiology that are relevant to explicating the nature of the selves. Note the plural. My reading of this literature is that there is not 1 multimodal self entity, but rather that there are separate and even competing selves within the organism. This perspective is missing in the authors presentation here. How does the broader literature of the field on the different forms of the self relate to the specific biological networks they have identified.

The authors conflation of the self and processing in the brain are illustrated here--

L86 The model reflects how the self  transmits and integrates information related to itself from the inside or outside, involving  three levels: interoceptive-processing, exteroceptive-processing, and 89 mental-self-processing[27].

What is the evidence that the SELF does this transmission and integration? I would think it obvious that it is not the self but neurological processing systems that are doing the processing. Otherwise we would have a interoceptive-processing self, exteroceptive-processing self, and mental-self-processing self. if we do then it become more imperative that these three selfs are well characterized and related to the psychological, philosophical and neuropsychiatric literature on forms of the self. There must be a congruence.

Figure 1 is fundamentally misleading. The implication with the arrows shown is that there are differences in Interoceptive, exteroceptive and mental processing that lead to "individual differences" that lead to "relevant brain regions".  Seems that the causal dynamics implied here by the horizontal arrows are fundamentally wrong.  Interoceptive, exteroceptive and mental processing systems do not lead to "individual differences." The model should be very different-- "relevant brain regions" --> self processing systems and that then there are "individual differences" in the tuning of these self processing systems (from biological variation, but more likely socialization) that lead to different types of BSC.

But just what are those types of BSC is never addressed in this article, in part because of the failure to consider the philosophy, psychology, neuropsychiatry of forms of self. Instead we are led to believe that variation in processing systems are in fact different forms of the self, but these different forms of BSC are not ever addressed.

In their three-level-self model, it is not established that these are 3 levels of one self as the authors often claim, as opposed to semi-autonomous entities. Clearly there is an information hierarchy in terms of biology to socialized features but this does not establish that they function as a hierarchy or that they are a self-hierarchy. Not well established that the self is strictly hierarchical in its information, nor in its forms (sub-selves" the 3 info processing systems). Clearly Mental level processes influence what is attended to at exteroceptive levels, so not a strict bottom up hierarchy. The authors go back and forth between a single versus multiple separate relatively autonomous selves. Which sometimes they assert: We propose that diverse experiences of BSC can be attributed to different modes of self-processing.

Just what are the primary modes?? This is where the tie into the broader literature is missing

But elsewhere it is clear they propose the three levels are always functioning together in one self.

They have not made a clear statement of just what the different self forms are and how related to underlying neural bases. How the hierarchy functions. How the aspects of the self presented here are relate to established neurological and evolutionary aspects of the self. i.e., other models that have a basis in neuropsychology and psychiatry and consciousness studies.

Review: commenting on the completeness of the review topic covered

As mentioned, the authors do not even contextualize sufficiently in the previous research they published on the neurological systems contributing to self processes and tentatively identify different forms of self. They need to begin  with a clear succinct review summary of the previous paper by two co-authors which should provide the context for this article.

Specific comments

L279 Mental self processing It is always associated with virtual mental-level 279 stimuli such as names, autobiographical memories, and trait words linked to the 280 self[89-91].

This implies that Mental self processing cannot be associated exclusively with some other features, certainly not true.

Line 285 concept of “face” is broader than explained and should be elaborated a little

Authors say: Following this section, we shall delve into how diverse levels of self-processing model work in synergy with each other to assert their pivotal influence in modulating the distinctive BSC.

But the following section does not provide such information

357 mental-self processing, as a  higher-level modal, is inherently associated with interoceptive-processing and multisensory processing, thereby advancing our comprehension of individual differences in BSC[115]

Does not make sense to me, something that is an  association advances comprehension of individual differences??,

L 394- Notably, convergent evidence suggests that the brain structure and function may 394 underlie individual differences in BSC[113,121

So the authors are proposing that people have different brain structures? and functions?? Seems like at most there is variation in degree of functions, but not different brain structures and functions

Authors say--On the other hand, the discrepancy in  empathy ability also leads to the consequence that the different effects of multisensory processing can be apparently observed.

discrepancy means lack of agreement-- There is a discrepancy when there is a difference between two things that should be alike.

What is being asserted here? that everyone should have the same empathy?? Somehow differences in empathy allows us to observe something about multisensory processing?

Table 1 seems to just repeat info from text verbatim and is unnecessary, I do not see that it contributes anything meaningful to the argumentation here

Move following?

Authors say: in line with our previously proposed theoretical hypothesis of the three-level-self model, we will review the literature pertinent to this framework, which allows conceptualizing the relationships between different levels of self-processing and individual differences in BSC [27].

OK

Authors also say: the goal of this review is to establish a comprehensive framework for elucidating the factors associated with self-processing that are linked to the individual differences in BSC, as well as bridging the unique perceptions of BSC with their  corresponding neural basis

I do not see anything that resembles a model or framework, much less comprehensive. What are the neural bases of these forms of self as opposed to the neural systems that inform the processes used in constructing the self? What is the evidence of "unique perceptions of BCS"-- if there is to be a science of BCS it must be nomothetic not ideosyncratic

Authors say: The third step is to consider the neural basis, which embodied individual differences in BSC, especially in the context of brain structure and function.

Seems to me that the differences described in aspects of the self are not about differences in brain structure and function-- we all have the same brain structures

I do not see a coherent statement about this "context of brain structure and function".

If there is a clear model, it seems that such findings could be expressed in a more pertinent Table with categories like

Level of Self Processing         Corresponding Neural Basis   Brain Structure and Function

Authors say: Based on relevant studies of brain structure and function, we will subsequently  provide an examination of the neural basis of individual differences in the three-level of self-processing, as well as the interaction among various levels

Authors say: This will involve searching the specific areas of the brain, their  activities, and how these relate to personal behavior and mind, thereby providing  valuable insights into the fundamental nature of human bodies, features, and consciousness

It seems to me that they ought to have a Table displaying what they promise, provide for each type/level of self

Table categories--

Specific areas of the brain      Brain  activities           Personal behavior and mind  Features of consciousness  Bodily experiences  Self-consciousness processing

and a table with these characteristics summarized

individual differences in BSC internal bodily signal perceptibility   multisensory processing principles    personal  traits            interaction modes

Authors say: To maintain coherence in multimodal self-representation, there is a requisite for interaction and synchronization across various processing levels [27,103].

Seems much of analytical psychology, psychodynamic psychology is addressing the lack of coherence across different forms of the self, with the control by the lower levels rather the apex.

329 The self as a multimodal entity is embedded in extensive discourses within psychology and philosophy [104,105]. Consequently, an integrated examination of 331 various levels of self-processing becomes crucial for comprehending the mechanisms that 332 underlie the formation of BSC [106].

Where is the integrated examination of  various levels of self-processing??

We are committed to conducting an in-depth examination at various 417 levels, with the intention of identifying the precise neural basis that underlie individual 418 differences in BSC. This will involve searching the specific areas of the brain, their 419 activities, and how these relate to personal behavior and mind, thereby providing 420 valuable insights into the fundamental nature of human bodies, features, and 421 consciousness (Table 1).

Seems like their offer to conduct an in-depth examination was at least partially done in the previous article that they should summarize here and provide new Tables as described above

Furthermore, Table 1 is not about specific brain areas, their activities, how they relate to personal behavior and mind. Rather it is about

Manifestations & Definitions

Main findings

Insights into

individual differences in BSC

Furthermore, some of the claims made, i.e., multimodal interaction provides reviews that do not seem focused on that. And some of the claims in that same section make claims about the causal aspects of empathy on lower lower self-processes that I do not see as being clearly substantiated in terms of causal directions

Some places make statements qualified by "may" since causal effects are not established, but here and elsewhere such needed qualifications are not made

three-level-self model presented above that allowed representing the relationship 508 between various bodily experiences and self-consciousness processing.

Evidence that the relationship is through self-consciousness processing??

BSC stems from the harmonious integration of interoceptive and exteroceptive signals, each contributing based on its anticipated precision [107,108].

But many people are not a harmonious integration, even if they want to be. Just what an anticipated precision is and how if assures a harmonious integration is unstated, and obviously non existent for many of those for whom a harmonious BCS does not always exist

  • Novelty: There is some novelty here but it is not made significant by being integrated into a broader model of the research it came from, nor the fields studying the self. Integration within the findings of the previous article and fields of studying the different forms of the self is necessary for the results provide an advancement of the current knowledge
  • Scope: This paper fits within the broadest scope of the journal.
  • Significance: This major lack significance because of lack of integration with relevant information from scientific studies on the self
  • Quality: Quality suffers for reasons stated above
  • Scientific Soundness: Only about 20% of the references are recent, past 5 years
  • Interest to the Readers: I think that this variation in aspects of bodily systems that process information is not very interesting or informative unless placed in the context of a broader model. If that were done I think it would arrive at far broader conclusions that would be interesting for the readership of the journal.
  • Overall Merit: I do not see much overall benefit in publishing this incomplete work. For it to advance current knowledge the current discussion needs to be placed in the context of a thorough presentation of the previous findings about the neurological bases of the self-processing systems. And these systems need to be placed in the context of other studies on forms, modes, types, levels of the self, not just variation in measures of self-processing mechanisms.
  • English Level: Grammatical problems are minor but frequent and should be corrected with a good editor.
  • Overall the paper is readable, comprehensible but there are a few sentences that I could not find a good understanding of (called out above)

Comments on the Quality of English Language
  • Grammatical problems are minor but frequent and should be corrected with a good editor.
  • Overall the paper is readable, comprehensible but there are a few sentences that I could not find a good understanding of (called out above)

Author Response

A brief summary 

Article provides a neuropsychology analysis of what the authors refer to as “Bodily self-consciousness” (BSC) concerned with awareness of one’s bodily states. They address the  individual differences in BSC in terms of the underlying factors and neural basis found as accounting for individual differences. They also assert to account for these individual differences in BSC using a three-level-self model, which includes interoceptive, exteroceptive, and mental-self-processing. But they also use this idea of levels of processing to discuss concepts of levels of self and modes of self and to equate the two levels.

Thanks for the reviewer’ s patience and pointing out the problems. We have provided a detailed response to your concerns below. I believe that under your suggestions, this paper has been significantly improved. 

The authors state: the goal of this review is to establish a comprehensive framework for elucidating the factors associated with self-processing that are linked to the individual differences in BSC, as well as bridging the unique perceptions of BSC with their corresponding neural basis. I do not see that such a comprehensive framework is presented nor the unique perceptions of BCS, or more importantly the different types of forms of BCS-- as opposed to information about variation in activity in neural processing components

Thanks for the reviewer’ s patience and pointing out the problems. We have given a detailed reply to this point below. I believe that under your suggestion, the paper has been greatly improved. 

Comment 1: Article: highlighting areas of weakness

L33, the very first sentence in the article highlights the overall weakness of the article:

 bodily self-consciousness (BSC) is so rooted in our daily life that it has increasingly become the focus in the fields of 35 philosophy, psychology and neuroscience[1-3].

But the references are just to brain science studies, nothing in philosophy and psychology or other relevant fields that study the self such as neuropsychiatry or consciousness studies.

Thank you for pointing this out. The initial sentence was due to the oversight in our expression. Indeed, the primary focus of our paper is on the cognitive neuroscience research associated with BSC, and we have accordingly revised the expression of Introduction. Please see page 1, lines 38-41:

“Bodily self-consciousness (BSC) is a fundamental aspect of human experience that refers to the subjective awareness of body as an aspect of self-consciousness experience. It has increasingly become the focus in the field of neuroscience (Blanke et al., 2015; Tsakiris et al., 2007; Schaller et al., 2021).”

Comment 2: Authors state that the individual differences of BSC can be traced back to the underlying neural basis of the brain.

I do not see how the article has made a direct presentation of this point, much less how even the different forms of self in the BSC are produced by specific underlying neural bases

Thank you for pointing this out. Our statement here is not rigorous, and it has been revised. Please see page 2, lines 91-95:

“The intricate relationship between our bodily experiences and the neural basis is a topic of immense interest in the field of neuroscience. As delving deeper into the diversity of BSC, the variation in BSC has piqued the curiosity of researchers, prompting a deeper investigation into its underlying neural basis. The revelation of individual differences in BSC has spurred a quest to uncover its biological roots.”

Comment 3: Line 56 Individual differences in BSC can be attributed to various aspects.

This presents a claim that I think the authors do not directly address, substantiate, but instead confuscate, mis-represent. Just what are the different types, kind, forms of BSC as opposed to variation in neural processes attributed roles self-related information.

The authors act here as if they are telling is what are these "individual differences" in self but in actuality they are presenting data about individual VARIATION in mechanisms involved in processing information. Variation on some measures of neurological processing, rather than anything that clearly presents what are the major forms of individual difference in self/selves.

Thank you for this comment. Following your suggestion, we have added relevant information in Table 3 and Table 1, such as the types of BSC and major forms of individual difference.

Table 3. Correlation Between Level of Self-Processing, Brain Areas, Brain networks, Brain Activities, Personal Behavior and Mind, Bodily Experiences, and Features of Consciousness

Level of self-processing

Specific areas of the brain

Brain network

Brain activities

Personal behavior and mind

Bodily experiences

Features of consciousness

Interoceptive-

processing

Insula (for the young) / ACC (for the elderly)

Salience network

Functional connectivity

Perception of heartbeat

Subjective perception of bodily awareness

Context-

dependent (for the young) / Context-

independent (for the elderly)

AIC

Salience network

Task-based activation

Performance accuracy on the interoceptive task

Interoceptive accuracy to bodily signals

Perceptibility and clarity of consciousness

Frontal lobe / BG

Default mode network / (/)

Functional connectivity

Self-reported sensitivity to interoceptive information

Sensitivity to internal bodily signals

Sensibility of consciousness

Visuospatial network

Visuospatial network

Functional connectivity

Maintaining normal cognitive function

Bodily representation ability

Degree of integration and association

Operculum / right anterior insula

Sensorimotor / salience network

Gray matter volume

Attention to either heart-beat timing or external note quality

Awareness of bodily and stress responses

Accuracy of interoceptive awareness

Exteroceptive-

processing

Insula / ACC

Salience network

Functional connectivity

Attention and cognitive control

Focus on sensory signals and bodily state

Integration of bottom-up and top-down sensory information

ACC

(bilateral caudal regions)

Salience network

Task-based activation

Integrating information over long timescales

Fine resolution of multisensory signals

Efficiency of multisensory processing

Parietal lobe

Default mode network

Task-based activation

Receiving input from sensory-specific cortices

Perception of multisensory input

Representation of multisensory modalities

Hippocampus

/

Task-based activation

Multisensory representation of the bodily self in space

Bodily self-location and out-of-body experience

Updatable and plastic

rlPFC / hippocampus

Executive Control Network / (/)

Task-based activation

Performing inference tasks related to Bayesian modeling

/

Hierarchical and complex

Mental-self-

processing

dmPFC / LPFC / TPJ

Default mode / frontoparietal network / (/)

Task-based activation

Self-relevance encoding during social affective process

Sensitive to bodily information related to significant others

Context-

dependent

ACC / vmPFC/ bilateral insula / right temporal pole

Salience / default mode / (/) / (/)

Task-based activation

Self-relevance encoding during social affective process

Attention to self-focused bodily information

Context-

independent

dmPFC / MTG / IFG / AG

Default mode / auditory / language / visual recognition network

Task-based activation

Processing semantics and integrating personal identity

Sense of identity distinction of bodily parts

Coherence and stability of self-

consciousness

vmPFC

Default mode network

Task-based activation

Caring about self-evaluation

Sensitive to bodily traits related to self-identity

Constructed by social identity

Multiple levels of self-processing

Insula / ACC

Salience network

Task-based activation

Experiences of complex social emotions

Compassion for physical pain

High-level cognitive processing

PMC / IPS

Motor / frontoparietal network

Task-based activation

Preferentially processing self-related signals

Self-location and identification

Global unity  and temporal continuity

Rostral PFC

Default mode network

Task-based activation

Comprehending of the others’ intentions

/

Emotion-

dependent

Right TPJ

/

Task-based activation

Comprehension of self-conscious emotions

/

Impacted by emotions

Note: ACC: anterior cingulate cortex, AIC: anterior insular cortex, BG: Basal ganglia, rlPFC: rostrolateral prefrontal cortex, dmPFC: dorsomedial prefrontal cortex, LPFC: lateral prefrontal cortex, TPJ: temporoparietal junction, vmPFC: ventromedial prefrontal cortex, MTG: middle temporal gyrus, IFG: inferior frontal gyrus, AG: angular gyrus, BSC: bodily self-consciousness, IPS: intraparietal sulcus, PMC: premotor cortex.

Table 1. The Manifestations of Individual Differences and Related Varied BSC

Manifestations of Individual Differences

Related Individual Differences in BSC

Internal bodily signal perceptibility

Interoceptive accuracy

·Higher interoceptive accuracy predicted narrower peripersonal space (Ardizzi & Ferri, 2018).

·Higher interoceptive accuracy indirectly correlated with stronger intensity of body illusions (Mai et al., 2018; Park et al., 2016).

Interoceptive sensibility

· Levels of interoceptive sensibility affected the vividness of RHI (Tsakiris, 2011).

· Interoceptive sensibility produced negative effects in body representation as individuals aged (Raimo, 2021).

· Females had higher interoceptive sensibility linked to larger insula volume, which contributes to various aspects of BSC (Longarzo et al., 2021).

Interoceptive awareness

· Males had greater interoceptive awareness, making them less prone to body illusions than females (Palmer & Tsakiris, 2018).

Multisensory processing principles

Bayesian causal inference principles (unisensory signal weighting)

·Individuals reliant on proprioception demonstrated enhanced body awareness and superior posture perception (Virtanen et al., 2022; Ehrsson, 2012).

· Individuals who assign higher weights for proprioceptive signals have a heightened sense of body part localization and are less prone to body illusions (Botan et al., 2021; Ernst & Banks, 2002).

Bayesian causal inference principles (multisensory processing)

·Bayesian CI computational principles explained the phenomenon that individuals with wider temporal binding windows experience stronger RHI (Litwin, 2020; Chancel et al., 2021, 2022).

Personal traits shaped by environment

Personality affected by sociocultural constructs

· East Asians are inclined to link the observed facial features with the social concept of “face” (Maister & Tsakiris, 2014).

· Japanese deeply ingrain formality and normativity into their traits, thereby overlooking their actual bodily sensations (Freedman et al., 2020).

·Americans tend to closely monitor their physical sensations because of their culture championing freedom and independence (Freedman et al., 2020).

Personal social identity

· White participants experienced more difficulty and took longer to mentally incorporate the image of a black rubber hand into their body schema (Lira et al., 2017).

· Women may be more prone to develop negative self-identification regarding their bodies (Kling et al., 2018).

Interaction modes

Interaction between low-level modals

·Individuals with higher interoceptive accuracy showed less decay in their prediction errors and lower exteroceptive precision, leading to reduced vividness in the RHI and whole-body illusion (Ainley et al., 2016).

Interaction between low and high-level modals

· Higher empathy correlates with superior bodily signal interpretation, aiding BSC evaluation (Mul et al., 2018; Fukushima et al., 2011).

· Individuals with higher empathy ability exhibited a more pronounced experience of RHI (Asai et al., 2011).

· Individuals with higher empathy ability experienced stronger self-localization drift and illusory self-identification (Mul et al., 2019).

· Individuals with higher empathy can switch perspectives between virtual bodies under synchronized mind-vision conditions (Heydrich et al., 2021).

Note: BSC: bodily self-consciousness, RHI: rubber hand illusion, Bayesian CI: Bayesian causal inference

Comment 4: Just what is the self, its individual biosocialized variant forms, is not spelled out. Just a lot about what are some of the systems that contribute to the information integrated in the organism that may contribute to self/selves.

Thank you for this comment. Following your suggestion, we have added the concept of the self to Introduction clearly. Please see page 1, lines 34-38:

“The self has long been employed in the research of psychology and neuroscience, whose integration can be perceived as extending from internal body to external natural and social environment (Park & Blanke, 2019; Qin et al., 2020). Self-consciousness is closely connected with the self and regarded as the feeling of self-experience, which is relied on the brain dynamic processing of diverse bodily signals (Adler et al., 2014).”

Comment 5: The authors refer to a previous study--  we here review previous findings on individual differences in BSC through a three-level-self model--but they do not actually review those previous findings and present them in a systematic way that contributes to this article or furthering the study of the neurological bases of the self and variation in modes of the self.

Thanks for this comment. Following your suggestion, we have added the review of previous findings that contributes to this paper, which proposed that there are multilevel of self-processing. Please see page 2, lines 66-80:

“Notably, our review is based on our previous theoretical model, which holds that there are three levels of self-processing, including interoceptive, exteroceptive, and mental-self-processing (Qin et al., 2020). The three-level-self model is used to explain how the brain integrates internal and external environmental information to reveal the relationship between body, environment and mental state in the self. The model embodies the mechanism of how external stimuli become self-related and thus integrate into the self, thereby extending the self beyond the physical body to encompass the environment. In addition, the results show that the common involvement of insula was found across all three levels of self-processing, which indicates that the integration of internal sensory signals may be the key to self-processing. In addition, cingulate cortex and anteromedial prefrontal cortex (amPFC) may play a role in combining non-physical external stimuli with body signals to form self-related pathways. The temporal-parietal junction (TPJ) plays an important role in integrating multisensory input and motion perception signals. These findings provide insights into how the brain processes complex information related to BSC.”

Comment 6: I have however, read a previous co-authored article by two of the authors (Qin, P.; Wang, M.; Northoff, G. 27 Qin, P.; Wang, M.; Northoff, G. Linking bodily, environmental and mental states in the self-A three-level model based on a meta-analysis. Neurosci Biobehav Rev. 2020, 115, 77–95. ), and I think the current paper requires a brief summary of the core findings from that previous paper in order to provide a neurological model that makes this paper a more comprehensible and a meaningful contribution.

As I elaborate below, I think a presentation (concise) of those previous finding is important for providing the context of what they want to present here.

Thanks for this comment. We have responded in our reply to Comment 5.

Comment 7: Authors say: The self as a multimodal entity is embedded in extensive discourses within psychology and philosophy[104,105

[104] Tajadura-Jiménez, A.; Grehl, S.; Tsakiris, M. The other in me: interpersonal multisensory stimulation changes the mental 775 representation of the self. PLoS One. 2012, 7, e40682.

[105] Desbordes, G. Self-related processing in mindfulness-based interventions. Neuron. 2019, 28, 312–316.

These are not considerations within psychology and philosophy. Indeed the paper has ignored a rich amount of information from these fields (psychology and philosophy) as well as psychiatry and psychobiology that are relevant to explicating the nature of the selves. Note the plural. My reading of this literature is that there is not 1 multimodal self entity, but rather that there are separate and even competing selves within the organism. This perspective is missing in the authors presentation here. How does the broader literature of the field on the different forms of the self relate to the specific biological networks they have identified.

Thanks for this comment. Our review is centered on the domain of cognitive neuroscience, thereby excluding evidence from philosophy and psychiatry. The original paper contained certain imprecise expressions that induced ambiguity, which have been revised. According to the previous findings, we have clearly illustrates the definition of self and unified the expression. Please see page 1, lines 34-36:

“The self has long been employed in the research of psychology and neuroscience, whose integration can be perceived as extending from internal body to external natural and social environment (Park & Blanke, 2019; Qin et al., 2020).”

Please see page 8, lines 377- 380:

“The definition of self is still controversial. For example, there is competition and integration among different levels of self inside the body (Tajadura-Jiménez et al., 2012; Desbordes, 2019). Consequently, this review attempts to explain the factors that associated with the individual differences in BSC from the perspective of multilevel self-processing (Legrand, 2010). ”

Comment 8: The authors conflation of the self and processing in the brain are illustrated here--

L86 The model reflects how the self  transmits and integrates information related to itself from the inside or outside, involving  three levels: interoceptive-processing, exteroceptive-processing, and 89 mental-self-processing[27].

What is the evidence that the SELF does this transmission and integration? I would think it obvious that it is not the self but neurological processing systems that are doing the processing. Otherwise we would have a interoceptive-processing self, exteroceptive-processing self, and mental-self-processing self. if we do then it become more imperative that these three selfs are well characterized and related to the psychological, philosophical and neuropsychiatric literature on forms of the self. There must be a congruence.

Thanks for pointing this out. Our expression resulted in the conceptual confusion, and it has been revised. Please see page 3, lines 122-125:

“The model reflects how the brain transmits and integrates information related to itself from the inside or outside, involving three levels: interoceptive-processing, exteroceptive-processing, and mental-self-processing (Qin et al., 2020).”

Comment 9: Figure 1 is fundamentally misleading. The implication with the arrows shown is that there are differences in Interoceptive, exteroceptive and mental processing that lead to "individual differences" that lead to "relevant brain regions".  Seems that the causal dynamics implied here by the horizontal arrows are fundamentally wrong.  Interoceptive, exteroceptive and mental processing systems do not lead to "individual differences." The model should be very different-- "relevant brain regions" --> self processing systems and that then there are "individual differences" in the tuning of these self processing systems (from biological variation, but more likely socialization) that lead to different types of BSC. 

Thanks for pointing this out. Following your suggestion, we have revised Figure 1.

(The picture can't be displayed here, please see the attachment for details.)

Figure 1. Specific Brain Regions among Various Levels of Self-processing and Their Related Manifestations of Individual Differences. Note: ACC: anterior cingulate cortex, AIC: anterior insular cortex, BG: Basal ganglia, rlPFC: rostrolateral prefrontal cortex, dmPFC: dorsomedial prefrontal cortex, LPFC: lateral prefrontal cortex, TPJ: temporoparietal junction, vmPFC: ventromedial prefrontal cortex, MTG: middle temporal gyrus, IFG: inferior frontal gyrus, AG: angular gyrus, IPS: intraparietal sulcus, PMC: premotor cortex

Comment 10: But just what are those types of BSC is never addressed in this article, in part because of the failure to consider the philosophy, psychology, neuropsychiatry of forms of self. Instead we are led to believe that variation in processing systems are in fact different forms of the self, but these different forms of BSC are not ever addressed.

Thanks for your suggestion. We acknowledged that the potential existence of other forms of self, but our focus in this study is limited to the self as it pertains to external information processing. We followed the previous three-level-self model, and the evidence we considered was also the evidence of cognitive neuroscience.Please see pages 1-2, lines 45-47:

“It was found that BSC stemmed from the multisensory integration and varied bodily experiences (including body ownership, self location, first-person perspective, and agency) (Blanke, 2012; Tsakiris, 2017).”

Comment 11: In their three-level-self model, it is not established that these are 3 levels of one self as the authors often claim, as opposed to semi-autonomous entities. Clearly there is an information hierarchy in terms of biology to socialized features but this does not establish that they function as a hierarchy or that they are a self-hierarchy. Not well established that the self is strictly hierarchical in its information, nor in its forms (sub-selves" the 3 info processing systems). Clearly Mental level processes influence what is attended to at exteroceptive levels, so not a strict bottom up hierarchy. The authors go back and forth between a single versus multiple separate relatively autonomous selves. Which sometimes they assert: We propose that diverse experiences of BSC can be attributed to different modes of self-processing. Just what are the primary modes?? This is where the tie into the broader literature is missing. 

Thanks for your suggestion. We acknowledge that the term “mode” or “modal” is inappropriate, and we have amended it to “level”, consistent with the previous research.

But elsewhere it is clear they propose the three levels are always functioning together in one self.

Thanks for pointing this out. Our expression is not clear and rigorous enough, leading to the conceptual misleading. In our opinion, it should be three layers of self-information processing and integration, which is helpful for self-perception.

They have not made a clear statement of just what the different self forms are and how related to underlying neural bases.

Thank you for this comment. Following your suggestion, we have added the information of what the different self forms are and how related to underlying neural bases. Please see page 3, lines 102- 111:

“We will discuss the neural basis of individual differences in BSC according to the main brain regions involved among three levels of self-processing. In the specific, the insula, anterior cingulate cortex (ACC) and frontal lobe are involved in all three levels (Chong et al., 2017; Smith et al., 2022; Han &Ma, 2014). The parietal lobe and hippocampus play a role in the integration and representation of sensory information under exteroception (Driver & Noesselt, 2008; Guterstam et al., 2015). The activity of TPJ and middle temporal gyrus (MTG) is related to mental-self-processing (D'Argembeau et al., 2014; Han &Ma, 2014). These main brain regions constitutes the neural basis of BSC, whose variations will aid in elucidating the individual differences in BSC.”

How the hierarchy functions.

Thank you for this comment. We added the information of how the hierarchy of self-processing functions. Please see page 1, lines 34-36:

“The self has long been employed in the research of psychology and neuroscience, whose integration can be perceived as extending from internal body to external natural and social environment (Park & Blanke, 2019; Qin et al., 2020).”

Please see page 2, lines 55-59:

“Previous studies have suggested that there are multilevel of self-processing, which refers to the processing of self-related body-environment information. In the context of self-processing, the perception of self-relevance needs to be progressive from low-level (interoception and exteroception) to high-level (mental-self) (Craig, 2009; Park & Blanke, 2019; Farmer et al., 2014).”

How the aspects of the self presented here are relate to established neurological and evolutionary aspects of the self. i.e., other models that have a basis in neuropsychology and psychiatry and consciousness studies.

Thank you for this comment. We have embodied the relevant information of how BSC integrates into three-level-self processing in Table 1, 2, and 3. We have showed the necessary information such as the Manifestations of Individual Differences and Related Varied BSC in three tables (Table 1), Corresponding Neural Basis of Self-processing (Table 2), the Features of Brain Regions and BSC to Each Self-processing Level (Table 3). Please see our response to Comment 21.

Review: commenting on the completeness of the review topic covered

As mentioned, the authors do not even contextualize sufficiently in the previous research they published on the neurological systems contributing to self processes and tentatively identify different forms of self. They need to begin with a clear succinct review summary of the previous paper by two co-authors which should provide the context for this article.

Thanks for your suggestion. This is the negligence of this review, and we have given a detailed reply about the previous research concerning the three-level-self model. Please see page 2, lines 66-80:

“Notably, our review is based on our previous theoretical model, which holds that there are three levels of self-processing, including interoceptive, exteroceptive, and mental-self-processing (Qin et al., 2020). The three-level-self model is used to explain how the brain integrates internal and external environmental information to reveal the relationship between body, environment and mental state in the self. The model embodies the mechanism of how external stimuli become self-related and thus integrate into the self, thereby extending the self beyond the physical body to encompass the environment. In addition, the results show that the common involvement of insula was found across all three levels of self-processing, which indicates that the integration of internal sensory signals may be the key to self-processing. In addition, cingulate cortex and anteromedial prefrontal cortex (amPFC) may play a role in combining non-physical external stimuli with body signals to form self-related pathways. The temporal-parietal junction (TPJ) plays an important role in integrating multisensory input and motion perception signals. These findings provide insights into how the brain processes complex information related to BSC.”

Specific comments:

Comment 12: L279 Mental self processing It is always associated with virtual mental-level 279 stimuli such as names, autobiographical memories, and trait words linked to the 280 self[89-91].

This implies that Mental self processing cannot be associated exclusively with some other features, certainly not true.

Thank you for pointing this out. We have revised this statement. Please see page 7, lines 319-323:

“It is often associated with virtual mental-level stimuli such as names, self identity, autobiographical memories, and trait words linked to the self (Lou et al., 2004; Amodio & Frith, 2006; Damasio, 2003). This level of self-processing involves brain regions like insula, ACC, PFC, and TPJ, affecting the connection between external irrelevant information and self, which will impact the self-processing at other levels.”

Comment 13: Line 285 concept of “face” is broader than explained and should be elaborated a little

Thank you for pointing this out. Following your suggestion, we have elaborated the concept of “face”. Please see page 7, lines 332-335:

“Particularly, in East Asian cultures, the concept of “face” is deeply rooted and refers to a person’s reputation, social status, and sense of pride or honor, which is an important aspect of interpersonal relationships and social interactions.”

Comment 14: Authors say: Following this section, we shall delve into how diverse levels of self-processing model work in synergy with each other to assert their pivotal influence in modulating the distinctive BSC.

But the following section does not provide such information

Thank you for pointing this out. We have revised the expression to clearly introduce the content of the next paragraph, making the context more fluent. Please see page 8, lines 370-373:

“To maintain the integration of multilevel self-representation, there is a requisite for interaction and synchronization across diverse processing levels (Qin et al., 2020; Moutoussis et al., 2014). Following this section, we shall delve into how diverse levels of self-processing interact with each other to assert their pivotal influence in modulating the distinctive BSC.”

Comment 15: 357 mental-self processing, as a  higher-level modal, is inherently associated with interoceptive-processing and multisensory processing, thereby advancing our comprehension of individual differences in BSC[115]

Does not make sense to me, something that is an  association advances comprehension of individual differences??,

Thank you for pointing this out. Our statement here has been revised. Please see page 9, lines 406-407:

“The differences of their interaction effect advance our comprehension of individual differences in BSC.”

Comment 16: L 394- Notably, convergent evidence suggests that the brain structure and function may 394 underlie individual differences in BSC[113,121

So the authors are proposing that people have different brain structures? and functions?? Seems like at most there is variation in degree of functions, but not different brain structures and functions

Thank you for pointing this out. We admitted that the brain structure remains consistent across individuals, variations in specific morphology is evident. Our paper subsequently reviewed the research evidence pertaining to these morphological differences.Please see page 9, lines 439-441:

“Notably, convergent evidence suggests that the variations in brain structure in terms of morphology and brain function may underlie individual differences in BSC (Tsakiris et al., 2010; Matuz-Budai et al., 2022).”

Comment 17: Authors say--On the other hand, the discrepancy in  empathy ability also leads to the consequence that the different effects of multisensory processing can be apparently observed.

discrepancy means lack of agreement-- There is a discrepancy when there is a difference between two things that should be alike.

What is being asserted here? that everyone should have the same empathy?? Somehow differences in empathy allows us to observe something about multisensory processing?

Thank you for pointing this out. We have revised the expression which was inaccurate. Please see page 9, lines 414-415:

“On the other hand, different levels of empathy ability lead to the different effects of multisensory processing.”

Comment 18: Table 1 seems to just repeat info from text verbatim and is unnecessary, I do not see that it contributes anything meaningful to the argumentation here

Move following?

Thank you for this comment. We acknowledged that the Table 1 was very long and seemed to just repeat info from text verbatim. Thus, we revised it below and showed the necessary information such as the Corresponding Neural Basis of Self-processing, the Features of Brain Regions and BSC to Each Self-processing Level, and Manifestations of Individual Differences and Related Varied BSC in three tables. Please see our response to Comment 21.

Comment 19: Authors say: in line with our previously proposed theoretical hypothesis of the three-level-self model, we will review the literature pertinent to this framework, which allows conceptualizing the relationships between different levels of self-processing and individual differences in BSC [27].

OK

Authors also say: the goal of this review is to establish a comprehensive framework for elucidating the factors associated with self-processing that are linked to the individual differences in BSC, as well as bridging the unique perceptions of BSC with their  corresponding neural basis

I do not see anything that resembles a model or framework, much less comprehensive.

Thank you for pointing this out. Our statement here has been revised. Please see page 3, lines 129-131:

“In sum, our goal is to synthesize the previous findings on individual differences in BSC from the perspective of the three-level-self model, aiming to enhance our comprehension of these individual differences and their corresponding neural basis.”

What are the neural bases of these forms of self as opposed to the neural systems that inform the processes used in constructing the self? What is the evidence of "unique perceptions of BCS"-- if there is to be a science of BCS it must be nomothetic not ideosyncratic

Thanks for your comment. Following your suggestion, we have illustrated what brain regions are involved in the three-level-self processing in Introduction that constitute the neural basis of BSC. Please see page 3, lines 102-111:

“We will discuss the neural basis of individual differences in BSC according to the main brain regions involved among three levels of self-processing. In the specific, the insula, anterior cingulate cortex (ACC) and frontal lobe are involved in all three levels (Chong et al., 2017; Smith et al., 2022; Han &Ma, 2014). The parietal lobe and hippocampus play a role in the integration and representation of sensory information under exteroception (Driver & Noesselt, 2008; Guterstam et al., 2015). The activity of TPJ and middle temporal gyrus (MTG) is related to mental-self-processing (D'Argembeau et al., 2014; Han &Ma, 2014). These main brain regions constitutes the neural basis of BSC, whose variations will aid in elucidating the individual differences in BSC.”

Comment 20: Authors say: The third step is to consider the neural basis, which embodied individual differences in BSC, especially in the context of brain structure and function.

Seems to me that the differences described in aspects of the self are not about differences in brain structure and function-- we all have the same brain structures

I do not see a coherent statement about this "context of brain structure and function".

Thanks for pointing this out. We made the ambiguity in our expression, resulting in the conceptual misleading. We have revised the related sentences related to the term brain structure. Please see page 2, lines 95-98:

“With the advent of neuroimaging technologies over the past decades, it has become feasible to gauge individual differences in brain structure in terms of morphology and brain function (Blanke et al., 2015; Serino et al., 2013; Olivé et al., 2015).”

Please see page 3, lines 140-142:

“The third step is to consider the neural basis, which embodied individual differences in BSC, especially in the context of brain structure in terms of morphology and brain function.”

Please see page 9, lines 439-441:

“Notably, convergent evidence suggests that the variations in brain structure in terms of morphology and brain function may underlie individual differences in BSC (Tsakiris et al., 2010; Matuz-Budai et al., 2022).”

Please see page 9, lines 442-444:

“On the one hand, as for the brain structure (morphology), individual differences in BSC can be explained by differences in cortical thickness (Matuz-Budai et al., 2022; Seghezzi et al., 2019).”

Please see page 10, lines 459-461:

“Based on relevant studies of brain structure (morphology) and function, we will subsequently provide a examination of the neural basis of individual differences in the three-level of self-processing, as well as the interaction among various levels.”

Please see page 18, lines 572-575:

“Secondly, at odds with studies on brain structure in terms of morphology and resting-state functional connectivity, we noted a lack of studies into task-based brain activation in building individual differences in BSC.”

Comment 21: If there is a clear model, it seems that such findings could be expressed in a more pertinent Table with categories like

Level of Self Processing         Corresponding Neural Basis   Brain Structure and Function

Thank you for your comment. Following your suggestion, we have created the table. Please see Table 2.

Table 2. Level of Self-processing and its Corresponding Neural Basis and Brain Structure and Function

Manifestations of Individual Differences 

Related Individual Differences in BSC

Internal bodily signal perceptibility

Interoceptive accuracy

· Higher interoceptive accuracy predicted narrower peripersonal space (Ardizzi & Ferri, 2018).

· Higher interoceptive accuracy indirectly correlated with stronger intensity of body illusions (Mai et al., 2018; Park et al., 2016).

Interoceptive sensibility

· Levels of interoceptive sensibility affected the vividness of RHI (Tsakiris, 2011).

· Interoceptive sensibility produced negative effects in body representation as individuals aged (Raimo, 2021).

· Females had higher interoceptive sensibility linked to larger insula volume, which contributes to various aspects of BSC (Longarzo et al., 2021).

Interoceptive awareness

· Males had greater interoceptive awareness, making them less prone to body illusions than females (Palmer & Tsakiris, 2018).

Multisensory processing principles

Bayesian causal inference principles (unisensory signal weighting)

· Individuals reliant on proprioception demonstrated enhanced body awareness and superior posture perception (Virtanen et al., 2022; Ehrsson, 2012).

· Individuals who assign higher weights for proprioceptive signals have a heightened sense of body part localization and are less prone to body illusions (Botan et al., 2021; Ernst & Banks, 2002).

Bayesian causal inference principles (multisensory processing)

· Bayesian CI computational principles explained the phenomenon that individuals with wider temporal binding windows experience stronger RHI (Litwin, 2020; Chancel et al., 2021, 2022).

Personal traits shaped by environment

Personality affected by sociocultural constructs

· East Asians are inclined to link the observed facial features with the social concept of “face” (Maister & Tsakiris, 2014).

· Japanese deeply ingrain formality and normativity into their traits, thereby overlooking their actual bodily sensations (Freedman et al., 2020).

· Americans tend to closely monitor their physical sensations because of their culture championing freedom and independence (Freedman et al., 2020).

Personal social identity

· White participants experienced more difficulty and took longer to mentally incorporate the image of a black rubber hand into their body schema (Lira et al., 2017).

· Women may be more prone to develop negative self-identification regarding their bodies (Kling et al., 2018).

Interaction modes

Interaction between low-level modals

· Individuals with higher interoceptive accuracy showed less decay in their prediction errors and lower exteroceptive precision, leading to reduced vividness in the RHI and whole-body illusion (Ainley et al., 2016).

Interaction between low and high-level modals

· Higher empathy correlates with superior bodily signal interpretation, aiding BSC evaluation (Mul et al., 2018; Fukushima et al., 2011).

· Individuals with higher empathy ability exhibited a more pronounced experience of RHI (Asai et al., 2011).

· Individuals with higher empathy ability experienced stronger self-localization drift and illusory self-identification (Mul et al., 2019).

· Individuals with higher empathy can switch perspectives between virtual bodies under synchronized mind-vision conditions (Heydrich et al., 2021).

Note: ACC: anterior cingulate cortex, AIC: anterior insular cortex, BG: Basal ganglia, rlPFC: rostrolateral prefrontal cortex, dmPFC: dorsomedial prefrontal cortex, LPFC: lateral prefrontal cortex, TPJ: temporoparietal junction, vmPFC: ventromedial prefrontal cortex, MTG: middle temporal gyrus, IFG: inferior frontal gyrus, AG: angular gyrus, DMN: default mode network, IPS: intraparietal sulcus, PMC: premotor cortex.

Comment 22: Authors say: Based on relevant studies of brain structure and function, we will subsequently  provide an examination of the neural basis of individual differences in the three-level of self-processing, as well as the interaction among various levels

Authors say: This will involve searching the specific areas of the brain, their  activities, and how these relate to personal behavior and mind, thereby providing  valuable insights into the fundamental nature of human bodies, features, and consciousness

It seems to me that they ought to have a Table displaying what they promise, provide for each type/level of self

Table categories--

Specific areas of the brain      Brain  activities           Personal behavior and mind  Features of consciousness  Bodily experiences  Self-consciousness processing

Thank you for your comment. Following your suggestion, we have created the table. Please see Table 3.

Table 3. Correlation Between Level of Self-Processing, Brain Areas, Brain Activities, Personal Behavior and Mind, Bodily Experiences, and Features of Consciousness

Level of self-processing

Specific areas of the brain

Brain network

Brain activities

Personal behavior and mind

Bodily experiences

Features of consciousness

Interoceptive-

processing

Insula (for the young) / ACC (for the elderly)

Salience network

Functional connectivity

Perception of heartbeat

Subjective perception of bodily awareness

Context-

dependent (for the young) / Context-

independent (for the elderly)

AIC

Salience network

Task-based activation

Performance accuracy on the interoceptive task

Interoceptive accuracy to bodily signals

Perceptibility and clarity of consciousness

Frontal lobe / BG

Default mode network / (/)

Functional connectivity

Self-reported sensitivity to interoceptive information

Sensitivity to internal bodily signals

Sensibility of consciousness

Visuospatial network

Visuospatial network

Functional connectivity

Maintaining normal cognitive function

Bodily representation ability

Degree of integration and association

Operculum / right anterior insula

Sensorimotor / salience network

Gray matter volume

Attention to either heart-beat timing or external note quality

Awareness of bodily and stress responses

Accuracy of interoceptive awareness

Exteroceptive-

processing

Insula / ACC

Salience network

Functional connectivity

Attention and cognitive control

Focus on sensory signals and bodily state

Integration of bottom-up and top-down sensory information

ACC

(bilateral caudal regions)

Salience network

Task-based activation

Integrating information over long timescales

Fine resolution of multisensory signals

Efficiency of multisensory processing

Parietal lobe

Default mode network

Task-based activation

Receiving input from sensory-specific cortices

Perception of multisensory input

Representation of multisensory modalities

Hippocampus

/

Task-based activation

Multisensory representation of the bodily self in space

Bodily self-location and out-of-body experience

Updatable and plastic

rlPFC / hippocampus

Executive Control Network / (/)

Task-based activation

Performing inference tasks related to Bayesian modeling

/

Hierarchical and complex

Mental-self-

processing

dmPFC / LPFC / TPJ

Default mode / frontoparietal network / (/)

Task-based activation

Self-relevance encoding during social affective process

Sensitive to bodily information related to significant others

Context-

dependent

ACC / vmPFC/ bilateral insula / right temporal pole

Salience / default mode / (/) / (/)

Task-based activation

Self-relevance encoding during social affective process

Attention to self-focused bodily information

Context-

independent

dmPFC / MTG / IFG / AG

Default mode / auditory / language / visual recognition network

Task-based activation

Processing semantics and integrating personal identity

Sense of identity distinction of bodily parts

Coherence and stability of self-

consciousness

vmPFC

Default mode network

Task-based activation

Caring about self-evaluation

Sensitive to bodily traits related to self-identity

Constructed by social identity

Multiple levels of self-processing

Insula / ACC

Salience network

Task-based activation

Experiences of complex social emotions

Compassion for physical pain

High-level cognitive processing

PMC / IPS

Motor / frontoparietal network

Task-based activation

Preferentially processing self-related signals

Self-location and identification

Global unity  and temporal continuity

Rostral PFC

Default mode network

Task-based activation

Comprehending of the others’ intentions

/

Emotion-

dependent

Right TPJ

/

Task-based activation

Comprehension of self-conscious emotions

/

Impacted by emotions

Note: ACC: anterior cingulate cortex, AIC: anterior insular cortex, BG: Basal ganglia, rlPFC: rostrolateral prefrontal cortex, dmPFC: dorsomedial prefrontal cortex, LPFC: lateral prefrontal cortex, TPJ: temporoparietal junction, vmPFC: ventromedial prefrontal cortex, MTG: middle temporal gyrus, IFG: inferior frontal gyrus, AG: angular gyrus, DMN: default mode network, BSC: bodily self-consciousness, IPS: intraparietal sulcus, PMC: premotor cortex.

and a table with these characteristics summarized

individual differences in BSC internal bodily signal perceptibility   multisensory processing principles    personal  traits            interaction modes

Thank you for your comment. Following your suggestion, we have created the table. Please see Table 1.

Table 1. The Manifestations of Individual Differences and Related Varied BSC

Manifestations of Individual Differences

Related Individual Differences in BSC

Internal bodily signal perceptibility

Interoceptive accuracy

·Higher interoceptive accuracy predicted narrower peripersonal space (Ardizzi & Ferri, 2018).

·Higher interoceptive accuracy indirectly correlated with stronger intensity of body illusions (Mai et al., 2018; Park et al., 2016).

Interoceptive sensibility

· Levels of interoceptive sensibility affected the vividness of RHI (Tsakiris, 2011).

· Interoceptive sensibility produced negative effects in body representation as individuals aged (Raimo, 2021).

· Females had higher interoceptive sensibility linked to larger insula volume, which contributes to various aspects of BSC (Longarzo et al., 2021).

Interoceptive awareness

· Males had greater interoceptive awareness, making them less prone to body illusions than females (Palmer & Tsakiris, 2018).

Multisensory processing principles

Bayesian causal inference principles (unisensory signal weighting)

·Individuals reliant on proprioception demonstrated enhanced body awareness and superior posture perception (Virtanen et al., 2022; Ehrsson, 2012).

· Individuals who assign higher weights for proprioceptive signals have a heightened sense of body part localization and are less prone to body illusions (Botan et al., 2021; Ernst & Banks, 2002).

Bayesian causal inference principles (multisensory processing)

·Bayesian CI computational principles explained the phenomenon that individuals with wider temporal binding windows experience stronger RHI (Litwin, 2020; Chancel et al., 2021, 2022).

Personal traits shaped by environment

Personality affected by sociocultural constructs

· East Asians are inclined to link the observed facial features with the social concept of “face” (Maister & Tsakiris, 2014).

· Japanese deeply ingrain formality and normativity into their traits, thereby overlooking their actual bodily sensations (Freedman et al., 2020).

·Americans tend to closely monitor their physical sensations because of their culture championing freedom and independence (Freedman et al., 2020).

Personal social identity

· White participants experienced more difficulty and took longer to mentally incorporate the image of a black rubber hand into their body schema (Lira et al., 2017).

· Women may be more prone to develop negative self-identification regarding their bodies (Kling et al., 2018).

Interaction modes

Interaction between low-level modals

·Individuals with higher interoceptive accuracy showed less decay in their prediction errors and lower exteroceptive precision, leading to reduced vividness in the RHI and whole-body illusion (Ainley et al., 2016).

Interaction between low and high-level modals

· Higher empathy correlates with superior bodily signal interpretation, aiding BSC evaluation (Mul et al., 2018; Fukushima et al., 2011).

· Individuals with higher empathy ability exhibited a more pronounced experience of RHI (Asai et al., 2011).

· Individuals with higher empathy ability experienced stronger self-localization drift and illusory self-identification (Mul et al., 2019).

· Individuals with higher empathy can switch perspectives between virtual bodies under synchronized mind-vision conditions (Heydrich et al., 2021).

Note: BSC: bodily self-consciousness, RHI: rubber hand illusion, Bayesian CI: Bayesian causal inference

Comment 23: Authors say: To maintain coherence in multimodal self-representation, there is a requisite for interaction and synchronization across various processing levels [27,103].

Seems much of analytical psychology, psychodynamic psychology is addressing the lack of coherence across different forms of the self, with the control by the lower levels rather the apex.

Thanks for pointing this out. We have revised the sentence that is not rigorous enough. Please see page 8, lines 370-371:

“To maintain the integration of multilevel self-representation, there is a requisite for interaction and synchronization across diverse processing levels (Qin et al., 2020; Moutoussis et al., 2014).”

Comment 24: 329 The self as a multimodal entity is embedded in extensive discourses within psychology and philosophy [104,105]. Consequently, an integrated examination of 331 various levels of self-processing becomes crucial for comprehending the mechanisms that 332 underlie the formation of BSC [106].

Where is the integrated examination of  various levels of self-processing??

Thank you for pointing this out. The integrated examination we were talking about actually refers to this review. We have revised the expression. Please see page 8, lines 378-380:

“Consequently, this review attempts to explain the factors that associated with the individual differences in BSC from the perspective of multilevel self-processing (Legrand, 2010). ”

Comment 25: We are committed to conducting an in-depth examination at various 417 levels, with the intention of identifying the precise neural basis that underlie individual 418 differences in BSC. This will involve searching the specific areas of the brain, their 419 activities, and how these relate to personal behavior and mind, thereby providing 420 valuable insights into the fundamental nature of human bodies, features, and 421 consciousness (Table 1).

Seems like their offer to conduct an in-depth examination was at least partially done in the previous article that they should summarize here and provide new Tables as described above

Furthermore, Table 1 is not about specific brain areas, their activities, how they relate to personal behavior and mind. Rather it is about

Manifestations & Definitions

Main findings

Insights into

individual differences in BSC

Thank you for this comment. We have recreated three tables and put them in the corresponding positions above. 

Comment 26: Furthermore, some of the claims made, i.e., multimodal interaction provides reviews that do not seem focused on that. And some of the claims in that same section make claims about the causal aspects of empathy on lower lower self-processes that I do not see as being clearly substantiated in terms of causal directions

Some places make statements qualified by "may" since causal effects are not established, but here and elsewhere such needed qualifications are not made

Thanks for your comment. Our expression has been revised. Please see page 9, lines 419-421:

“Similarly, Mul et al. (2019) noted a correlation between higher empathy ability and increased susceptibility to body illusions when exposed to synchronized visual-tactile stimuli, which exhibited stronger self-localization drift and illusory self-identification.”

Please see page 9, lines 423-427:

“Heydrich et al. (2021) found that under the synchronized mind-vision conditions, individuals with higher empathy ability were able to swiftly switch perspectives between the virtual bodies of self and those of others, which relied heavily on the capacity to perceive others’ situations.”

Comment 27: three-level-self model presented above that allowed representing the relationship 508 between various bodily experiences and self-consciousness processing.

Evidence that the relationship is through self-consciousness processing??

Thanks for your comment. We have carefully revised the expression. Please see page 18, lines 567-569:

“Our review comprehensively contemplated individual differences in BSC, guided by a three-level-self model presented above that allowed representing the relationship between various bodily experiences and self-processing.”

Comment 28: BSC stems from the harmonious integration of interoceptive and exteroceptive signals, each contributing based on its anticipated precision [107,108].

But many people are not a harmonious integration, even if they want to be. Just what an anticipated precision is and how if assures a harmonious integration is unstated, and obviously non existent for many of those for whom a harmonious BCS does not always exist

Thanks for your comment. Our expression is not rigorous enough, leading to a absolute context, and it has been revised. Please see page 8, lines 383-384:

“BSC stems from the integration of interoceptive and exteroceptive signals, each contributing based on its relative precision (Banellis & Cruse, 2020; Seth, 2013).” 

Novelty: There is some novelty here but it is not made significant by being integrated into a broader model of the research it came from, nor the fields studying the self. Integration within the findings of the previous article and fields of studying the different forms of the self is necessary for the results provide an advancement of the current knowledge

Scope: This paper fits within the broadest scope of the journal.

Significance: This major lack significance because of lack of integration with relevant information from scientific studies on the self

Quality: Quality suffers for reasons stated above

Scientific Soundness: Only about 20% of the references are recent, past 5 years

Interest to the Readers: I think that this variation in aspects of bodily systems that process information is not very interesting or informative unless placed in the context of a broader model. If that were done I think it would arrive at far broader conclusions that would be interesting for the readership of the journal.

Overall Merit: I do not see much overall benefit in publishing this incomplete work. For it to advance current knowledge the current discussion needs to be placed in the context of a thorough presentation of the previous findings about the neurological bases of the self-processing systems. And these systems need to be placed in the context of other studies on forms, modes, types, levels of the self, not just variation in measures of self-processing mechanisms.

English Level: Grammatical problems are minor but frequent and should be corrected with a good editor.

Overall the paper is readable, comprehensible but there are a few sentences that I could not find a good understanding of (called out above)

We have added some references in recent five years.We highly appreciate the reviewers’ insightful comments and have addressed each suggestion in our revision.

Round 2

Reviewer 1 Report

Comments and Suggestions for Authors

Thank you for these revisions.

Author Response

Comment: Thank you for these revisions.

Response: We appreciate the reviewer’s comment.

Reviewer 2 Report

Comments and Suggestions for Authors

Figure 1 as presented in the Cover letter makes sense to me.

However in the paper Figure 1  is organized differently, I think as in the original, and does not make sense to me. Seems there was an error, the Figure in the paper was not corrected as indicated in the cove letter

A diagram of different differences

Description automatically generated

448-449 Notably, convergent evidence suggests that the variations in brain structure in 449 terms of morphology and brain function may underlie individual differences in BSC

Again I do not see that the authors have presented evidence that there are "variations in brain function" . Differences in the degree of connectivity, yes, but this is not a difference in function.

Table 3

Extraneous text in second column. Seems that there is missing text in a number of the cells for areas of exteroceptive processing and multilevel processing. Presentation in the cover letter seems fine but in the actual paper there is scrambled nformation

Comments on the Quality of English Language

still has some grammatical errors

Author Response

Authors’ Response to Reviewer Comments

We are truly grateful to the editors for coordinating the review of our manuscript and to the reviewers for their insightful and constructive comments. By making thorough and careful revisions to the manuscript based on the reviewer’ comments, we believe that this manuscript has been greatly improved.

Our point-by-point responses to the comments of the reviewer are listed below. The changes in the revised manuscript are marked in blue.

Reviewer #2:

Comment 1: Figure 1 as presented in the Cover letter makes sense to me.

However in the paper Figure 1  is organized differently, I think as in the original, and does not make sense to me. Seems there was an error, the Figure in the paper was not corrected as indicated in the cove letter

Response 1: Thank you for pointing this out. We have corrected the Figure 1 in the paper.

(Figure 1 cannot be displayed here, please check the supplementary files.)

Figure 1. Specific Brain Regions among Various Levels of Self-processing and Their Related Manifestations of Individual Differences. Note: ACC: anterior cingulate cortex, AIC: anterior insular cortex, BG: Basal ganglia, rlPFC: rostrolateral prefrontal cortex, MPFC: medial prefrontal cortex, LPFC: lateral prefrontal cortex, IFG: inferior frontal gyrus, MTG: middle temporal gyrus, TPJ: temporoparietal junction, AG: angular gyrus, PMC: premotor cortex, IPS: intraparietal sulcus.

Comment 2: 448-449 Notably, convergent evidence suggests that the variations in brain structure in 449 terms of morphology and brain function may underlie individual differences in BSC

Again I do not see that the authors have presented evidence that there are "variations in brain function" . Differences in the degree of connectivity, yes, but this is not a difference in function.

Response 2: Thanks for your comment. Following your suggestion, we have revised the expression that was not clear and rigorous enough, which led to the conceptual misleading. Please see page 3, lines 100- 103:

“Furthermore, the examination of the variations in the degree of the brain functional connectivity and task-based activation has also proven beneficial for probing individual differences of BSC, encompassing the neural correlates of the first-person perspective (Candia-Rivera et al., 2021).”

Please see page 3, lines 140- 142:

“The third step is to consider the neural basis, which embodied individual differences in BSC, especially in the context of brain structure in terms of morphology, brain functional connectivity and task-based activation.”

Please see page 12, lines 477- 479:

“Notably, convergent evidence suggests that the variations in brain structure in terms of morphology and degree of brain functional connectivity and task-based activation may underlie individual differences in BSC (Tsakiris et al., 2010; Matuz-Budai et al., 2022).”

Please see page 12, lines 497- 500:

“Based on relevant studies of brain structure (morphology), functional connectivity, and task-based activation, we will subsequently provide a examination of the neural basis of individual differences in the three-level of self-processing, as well as the interaction among various levels.”

Comment 3: Table 3

Extraneous text in second column. Seems that there is missing text in a number of the cells for areas of exteroceptive processing and multilevel processing. Presentation in the cover letter seems fine but in the actual paper there is scrambled information

Response 3: Thank you for pointing this out. Following your suggestion, we have renewed and presented the complete Table 3 in the response here and the paper, to ensure that the text in second column can be presented normally.

Table 3. Correlation Between Level of Self-Processing, Brain Areas, Brain Activities, Personal Behavior and Mind, Bodily Experiences, and Features of Consciousness.

Level of self-processing

Specific areas of the brain

Brain network

Brain activities

Personal behavior and mind

Bodily experiences

Features of consciousness

Interoceptive-

processing

Insula (for the young) / ACC (for the elderly)

Salience network

Functional connectivity

Perception of heartbeat

Subjective perception of bodily awareness

Context- dependent (for the young) / Context- independent (for the elderly)

AIC

Salience network

Task-based activation

Performance accuracy on the interoceptive task

Interoceptive accuracy to bodily signals

Perceptibility and clarity of consciousness

Frontal lobe / BG

Default mode network / (/)

Functional connectivity

Self-reported sensitivity to interoceptive information

Sensitivity to internal bodily signals

Sensibility of consciousness

Visuospatial network

Visuospatial network

Functional connectivity

Maintaining normal cognitive function

Bodily representation ability

Degree of integration and association

Operculum / right anterior insula

Sensorimotor / salience network

Gray matter volume

Attention to either heart-beat timing or external note quality

Awareness of bodily and stress responses

Accuracy of interoceptive awareness

Exteroceptive-

processing

Right PFC / right inferior frontoparietal cortices

Frontoparietal network

Task-based activation

Self-face recognition and facial expression processing

Sense of identification and ownership with one’s own body

Stable, self-bias, and continually updated

Insula / ACC

Salience network

Functional connectivity

Attention and cognitive control

Focus on sensory signals and bodily state

Integration of bottom-up and top-down sensory information

ACC (bilateral caudal regions)

Salience network

Task-based activation

Integrating information over long timescales

Fine resolution of multisensory signals

Efficiency of multisensory processing

Parietal lobe

Default mode network

Task-based activation

Receiving input from sensory-specific cortices

Perception of multisensory input

Representation of multisensory modalities

Hippocampus

/

Task-based activation

Multisensory representation of the bodily self in space

Bodily self-location and out-of-body experience

Updatable and plastic

rlPFC / hippocampus

Executive Control Network / (/)

Task-based activation

Performing inference tasks related to Bayesian modeling

/

Hierarchical and complex

Mental-self-

processing

dmPFC / LPFC / TPJ

Default mode / frontoparietal network / (/)

Task-based activation

Self-relevance encoding during social affective process

Sensitive to bodily information related to significant others

Context- dependent

ACC / vmPFC/ bilateral insula / right temporal pole

Salience / default mode / (/) / (/)

Task-based activation

Self-relevance encoding during social affective process

Attention to self-focused bodily information

Context- independent

dmPFC / MTG / IFG / AG

Default mode / auditory / language / visual recognition network

Task-based activation

Processing semantics and integrating personal identity

Sense of identity distinction of bodily parts

Coherence and stability of self- consciousness

vmPFC

Default mode network

Task-based activation

Caring about self-evaluation

Sensitive to bodily traits related to self-identity

Constructed by social identity

Multiple levels of self-processing

Insula / ACC

Salience network

Task-based activation

Experiences of complex social emotions

Compassion for physical pain

High-level cognitive processing

PMC / IPS

Motor / frontoparietal network

Task-based activation

Preferentially processing self-related signals

Self-location and identification

Global unity  and temporal continuity

Rostral PFC

Default mode network

Task-based activation

Comprehending of the others’ intentions

/

Emotion- dependent

Right TPJ

/

Task-based activation

Comprehension of self-conscious emotions

/

Impacted by emotions

Note: ACC: anterior cingulate cortex, AIC: anterior insular cortex, BG: Basal ganglia, PFC: prefrontal cortex, rlPFC: rostrolateral prefrontal cortex, dmPFC: dorsomedial prefrontal cortex, LPFC: lateral prefrontal cortex, TPJ: temporoparietal junction, vmPFC: ventromedial prefrontal cortex, MTG: middle temporal gyrus, IFG: inferior frontal gyrus, AG: angular gyrus, PMC: premotor cortex, IPS: intraparietal sulcus.

Comment 4: Table 2 The Brain Structure and Function column has been collapsed, lost the formatting

Response 4: Thanks for pointing this out. Following your suggestion, we have presented the complete Table 2 in the response here and the paper, to ensure that the text in The Brain Structure and Function column can be presented normally.

Table 2. Level of Self-processing and its Corresponding Neural Basis and Brain Structure and Function.

Level of Self-processing

Corresponding Neural Basis

Brain Structure and Function

Interoceptive-

processing

· Level of interoceptive accuracy is indirectly linked to BSC via ACC and insula connectivity (Ueno et al., 2020; Chong et al., 2017).

· Wang et al. (2019) found that task-based activation of the AIC predicted variations in interoceptive accuracy.

· A negative correlation was found between interoceptive sensitivity and functional connectivity in frontal lobe and BG (Smith et al., 2022).

· The decline in bodily representation ability with age among those with higher interoceptive sensitivity may be due to altered brain network function like visuospatial network (Betzel et al., 2014; Bagarinao et al., 2019).

· A strong correlation existed between interoceptive awareness and regional gray matter volume in operculum and right anterior insula (Pollatos et al., 2007; Critchley et al., 2004; Caseras et al., 2013).

· Functional connectivity in ACC and insula

· Task-based activation of AIC

· Functional connectivity in frontal lobe and BG

· Functional connectivity in visuospatial network

· Gray matter volume in operculum and right anterior insula

Exteroceptive-

processing

· The right PFC and right inferior frontoparietal cortices involved in the self-face recognition (Keenan et al., 2000; Morita et al., 2017).

· Both insula and ACC are essential for filtering and processing sensory information (Menon & Uddin, 2010; Yantis, 2008).

· There is a strong correlation between the bilateral caudal regions of the ACC within the frontal cortex and the width of the temporal binding window (Johnston et al., 2022).

· Parietal lobe integrates sensory information, aiding in coordinating behavioral responses (Driver & Noesselt, 2008).

· The salience network can process multisensory stimuli and modulate other core networks like the attention and cognitive control networks (Menon & Uddin, 2010; Yantis, 2008).

· Activity in hippocampus reflected the multisensory representation of the bodily self in space (Guterstam et al., 2015).

· rlPFC and hippocampus showed increased activation for Bayesian modeling inference tasks (Katayama et al., 2024).

· Task-based activation of the right PFC and right inferior frontoparietal cortices

· Functional connectivity in insula and ACC

· Task-based activation of the bilateral caudal regions of ACC

· Task-based activation of parietal lobe

· Task-based activation of salience, attention and cognitive control networks

· Task-based activation of rlPFC and hippocampus

Mental-self-

processing

· East Asians typically show increased neural activity in the dmPFC, LPFC, and TPJ associated with their interdependent self-construal, while Westerners often exhibit greater activation in the ACC, vmPFC, bilateral insula, and right temporal pole related to their independent self-construal (Han & Ma, 2014; Chiao et al., 2009).

· Self-consciousness and self-identity recognition rely on specific brain regions for regulation, including dmPFC, MTG, IFG, and AG, which involved in processing semantics and integrating personal identity with past experiences (D'Argembeau et al., 2014; Binder et al., 2009; Price, 2010).

· The vmPFC, part of the DMN, plays a role in processing self-relevant information and social cognitive tasks, as well as maintaining coherence of self-concept (Elder et al., 2023; Amodio & Frith, 2006; Menon & Uddin, 2010).

· The precuneus, a key component of DMN, is vital for linking the BSC to socially-related emotional information, as well as mediating self-referential processing (Lanius et al., 2015; Cabanis et al., 2013).

· Task-based activation of dmPFC, LPFC, and TPJ

· Task-based activation of ACC, vmPFC, bilateral insula, and right temporal pole

· Task-based activation of dmPFC, MTG, IFG, and AG

· Task-based activation of vmPFC

· Task-based activation of precuneus

Multiple levels of self-processing

· The integrated body system harmoniously blends interoception and exteroception, with the IPS and PMC contributing to this interaction (Park & Blanke, 2019).

· The insula and ACC play a crucial role in complex social emotions such as empathy, with activation in insula positively correlated with empathy ability (Immordino-Yang et al., 2009; Singer et al., 2004).

· The awareness and comprehending of the intentions of others was shown to involve rostral PFC (Seitz et al., 2006).

· The right TPJ is responsible for comprehension of self-conscious emotions like empathy (Lavarco et al., 2022).

· Task-based activation of IPS and PMC

· Task-based activation of insula and ACC

· Task-based activation of PFC

· Task-based activation of TPJ

Note: BSC: bodily self-consciousness, ACC: anterior cingulate cortex, AIC: anterior insular cortex, BG: Basal ganglia, rlPFC: rostrolateral prefrontal cortex, dmPFC: dorsomedial prefrontal cortex, LPFC: lateral prefrontal cortex, TPJ: temporoparietal junction, vmPFC: ventromedial prefrontal cortex, MTG: middle temporal gyrus, IFG: inferior frontal gyrus, AG: angular gyrus, DMN: default mode network, IPS: intraparietal sulcus, PMC: premotor cortex.
